# CG-SSL: Concept-Guided Self-Supervised Learning

**Sara Atito**[*1,2]        **Josef Kittler**[2]        **Imran Razzak**[3]        **Muhammad Awais**[1,2]

[1] Surrey Institute for People-Centred AI, University of Surrey, UK

[2]Centre for Vision, Speech and Signal Processing (CVSSP), University of Surrey, UK

[3] Mohamed bin Zayed University of Artificial Intelligence (MBZUAI), Abu Dhabi, UAE

sara.atito@surrey.ac.uk,imran.razzak@mbzuai.ac.ae
j.kittler@surrey.ac.uk,muhammad.awais@surrey.ac.uk

## Abstract

Humans understand visual scenes by first capturing a global impression and then refining this understanding into distinct, object-like components. Inspired by this process, we introduce **C**oncept-**G**uided **S**elf-**S**upervised **L**earning (CG-SSL), a novel framework that brings structure and interpretability to representation learning through a curriculum of three training phases: (1) global scene encoding, (2) discovery of visual concepts via tokenised cross-attention, and (3) alignment of these concepts across views. Unlike traditional SSL methods, which simply enforce similarity between multiple augmented views of the same image, CG-SSL accounts for the fact that these views may highlight different parts of an object or scene. To address this, our method establishes explicit correspondences between views and aligns the representations of meaningful image regions. At its core, CG-SSL augments standard SSL with a lightweight decoder that learns and refines concept tokens via cross-attention with patch features. The concept tokens are trained using masked concept distillation and a feature-space reconstruction objective. A final alignment stage enforces view consistency by geometrically matching concept regions under heavy augmentation, enabling more compact, robust, and disentangled representations of scene regions. Across multiple backbone sizes, CG-SSL achieves state-of-the-art results on image segmentation benchmarks using $k$-NN and linear probes, substantially outperforming prior methods and approaching, or even surpassing, the performance of leading SSL models trained on over $100\times$ more data. Code and pretrained models will be released.

## 1  Introduction

Deep neural networks have achieved huge success across a wide range of computer vision tasks, but this success has come at the cost of increasingly large and expensive labeled datasets. Self-supervised learning (SSL) offers a scalable alternative, enabling models to learn generalisable representations from unlabeled data. SSL has shown promise not only in mainstream tasks such as image classification and segmentation [1, 2, 3], but also in domains where annotations are limited or costly, such as medical imaging [4, 5, 6], satellite imagery [7], underwater exploration [8], and beyond.

Within the SSL landscape, two families of approaches have proven particularly effective. Generative methods, such as masked image modeling (MIM) [9, 10, 11, 12, 13], train models to reconstruct occluded patches of input images. These methods are particularly effective at learning spatially grounded features and fine-grained textures. However, they often fall short when it comes to capturing high-level semantic information, typically underperforming on transfer tasks that require robust, discriminative features [14]. In practice, such models require extensive fine-tuning and

---

[*]Project Lead.

39th Conference on Neural Information Processing Systems (NeurIPS 2025).

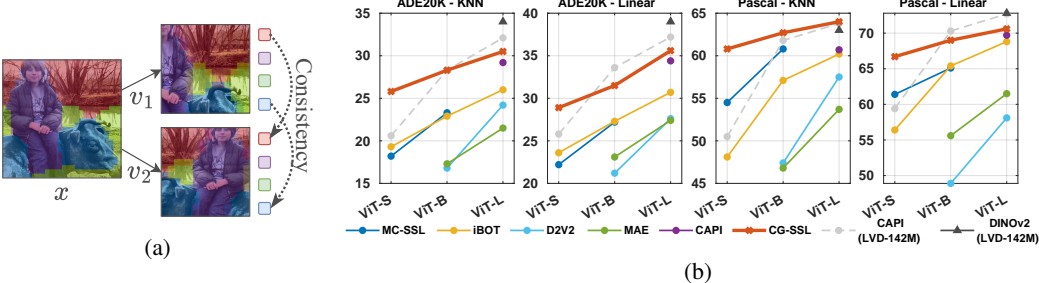

(a)

(b)

Figure 1: (a) CG-SSL overview: The model segments an image into $N$ coherent "concept" regions. Original image is passed through the teacher to obtain attention masks. Using the known geometry of the crops, these masks are accurately mapped to augmented views $v_1, v_2$. Tokens from each region are aggregated and consistency loss is applied between corresponding regions. (b) We evaluate frozen representations using $k$-NN and linear probes. CG-SSL outperforms prior baselines, scales well with model size, and in some cases surpasses models pre-trained on much larger datasets.

fail to generalise well with simple linear probes. In contrast, invariance-based methods [15, 16, 17, 18, 19] focus on aligning the representations of different augmented views of the same image. These approaches produce features that are generally more transferable and better suited to global classification tasks. Nonetheless, they rely on a critical assumption that the applied augmentations do not alter the semantic content of the scene. In cluttered or complex visual environments, this assumption frequently breaks down. Moreover, by compressing the image into a single vector representation, these methods tend to discard spatial information, making them poorly suited for tasks that require localisation or fine-grained reasoning, such as object detection and segmentation.

Recent works have attempted to bridge the gap between these paradigms. Methods like iBoT [16, 20] and DINOv2 [21] integrate token-level predictions and masked modeling objectives to enhance spatial awareness. While these models demonstrate strong performance and scalability, they continue to treat image patches largely as independent units and do not explicitly encourage the formation of coherent visual entities. As such, they fall short of modeling the structured, hierarchical nature of visual perception observed in humans, where coherent regions are grouped [22, 23].

Finally, recent advances in region- and object-centric SSL [24, 25, 26, 27] have made progress toward structured visual representations. However, many of these methods depend on pre-trained models or heuristic objectives, which limits their scalability and hinders end-to-end learning.

To address these limitations, we propose Concept-Guided Self-Supervised Learning (CG-SSL), a framework that explicitly discovers, tracks, and aligns coherent visual regions "referred to as concepts" across augmented views. Rather than relying on implicit alignment through crop overlap, CG-SSL learns a set of concept tokens dedicated embeddings that attend to image patches via cross-attention. These tokens are refined using two auxiliary objectives: a masked concept distillation loss that promotes consistency, and a feature-space reconstruction loss that encourages diversity and completeness. In the final phase, CG-SSL tracks these discovered concepts across augmented views by leveraging known geometric transformations. Using a reference image (a lightly augmented version of the original - no cropping) as a stable anchor, we project attention masks into each augmented view and aggregate the corresponding features to enforce cross-view consistency. This structured alignment enables CG-SSL to learn stable and interpretable representations without any supervision.

By combining global understanding with region-level structure and view-consistent alignment, CG-SSL moves beyond existing limitations in SSL by producing representations that are not only transferable but also interpretable and spatially coherent. We demonstrate that CG-SSL achieves state-of-the-art (SOTA) performance on image segmentation benchmarks using only simple probes, and approaches or surpasses models trained on orders of magnitude more data.

## 2  Related Works

SSL methods largely fall into two broad families: generative approaches that reconstruct input signals, and invariance-based approaches that learn representations by enforcing consistency across views.

As our work aligns more closely with the latter, we focus here on invariance-based methods, which we organise into three levels of granularity: image-level, patch-level, and region-level.

**Image-Level SSL.** A key milestone in SSL was instance discrimination [28], where each image is treated as its own class. This formulation paved the way for contrastive approaches such as InfoNCE [29], SimCLR [19], and MoCo [17], which encourage agreement between views of the same image while contrasting them against others. These methods achieved solid results on downstream tasks, establishing cross-view consistency as a central principle in SSL. Subsequent work focused on preventing collapse. Negative sampling [30] remains one solution, though it often relies on large batch sizes. Alternatives include momentum encoders[15], clustering-based objectives [31, 18], and entropy-based regularisation [32], all of which stabilise learning without explicit negatives.

While these methods vary in their learning objectives, they share a common assumption: the image is treated as a holistic unit. However, this global perspective overlooks the rich spatial structure of natural scenes. To address this, recent efforts have shifted toward learning at the patch or region level, where spatial relationships and part-whole semantics play a central role.

**Patch-Level SSL.** Pixel- and patch-level SSL extends instance discrimination from global image embeddings to local features within the spatial grid [33, 34, 35, 36, 16, 20, 21]. These methods aim to learn dense representations by enforcing local consistency within and/or across augmented views. The primary distinction among them lies in how positive patch correspondences are defined. Some works [33, 28] use contrastive losses (e.g., InfoNCE) where matching pairs are drawn from the same spatial location in the student and teacher outputs, while others [35, 36] rely on nearest-neighbor patch alignment in feature space across views. Another class of methods, including iBOT [20] and DINOv2 [21], extend global cross-entropy objectives to the patch level, matching representations of masked and unmasked views in a studentteacher setup.

These patch-level objectives yield strong performance on dense prediction tasks. However, they still treat each patch independently, without explicit modeling of spatial coherence across regions. As a result, they may fail to capture structured patterns like compositional regions, which motivates the transition to region-level methods that learn to group patches into coherent regions/concepts.

**Region-/Concept-Level SSL.** While image- and patch-level SSL has seen great success, recent work has shifted toward learning structured visual representations by grouping regions or objects within images. However, many of these methods are primarily designed for unsupervised segmentation or object discovery, rather than general-purpose SSL. For example, PiCIE [37], STEGO [38], and SegDiscover [39] cluster dense pixel features to enable semantic segmentation, often relying on hand-tuned spatial priors or post-processing. Other approaches like SlotCon [24, 25], CrOC [26], and CrIBo [27] aim to learn object-centric representations, but often depend on clustering or external region grouping, limiting scalability.

In contrast, CG-SSL is designed as a general-purpose self-supervised framework. It learns to discover and align coherent visual concepts in a fully end-to-end manner, without pretraining, offline clustering, or handcrafted proposals. Concept tokens emerge via attention-based grouping, and cross-view consistency is enforced through geometric alignment of discovered regions.

## 3 Methodology

We propose CG-SSL framework, which follows a three-phase curriculum: (1) capturing the global image gist, (2) segmenting the scene into coherent regions, and (3) aligning these regions across viewpoints. Each phase builds on the previous one, progressively enriching the learned representations. Training is staged, with each phase introducing new objectives while retaining earlier ones which enables the model to develop from broad understanding to fine-grained, aligned representations, resulting in robust and interpretable SSL. An overview and results are shown in Figure 1.

### 3.1 Phase 1: Global Representation Learning

Human perception begins with an immediate grasp of the global structure of a scene [40, 41]. Cognitive psychology refers to this as global precedence, where low spatial frequency information helps the visual system quickly form a "gist" of a scene before identifying individual components.

The content of a gist usually includes a conceptual understanding of a scene, e.g. birthday party, the spatial layout of the environment, and the identity of a few objects [42].

In machine vision, SSL has paralleled this phase through invariance-based methods, which aim to learn global representations by making the model recognise that different views of the same image correspond to the same underlying content. While this first phase of global feature learning is crucial, it is also a well-established and heavily explored subfield of SSL. Rather than proposing a new formulation, we adopt iBoT [20], a successor to DINO [18], because of its strong empirical results and architectural simplicity. iBoT aligns both image-level cls token and patch-level embeddings produced by a student­teacher pair of Vision Transformers (ViTs).

Let $\mathcal{E}_s, \mathcal{E}_t$ denote the student and teacher encoders, and $\mathcal{P}_s^{[\text{cls}]}$, $\mathcal{P}_t^{[\text{cls}]}$, $\mathcal{P}_s^{[\text{patch}]}$, $\mathcal{P}_t^{[\text{patch}]}$ their respective projection heads for the cls and patch streams. From a single image $x \in \mathbb{R}^{H \times W}$ we draw two stochastic augmentations $v_1, v_2 \sim \mathcal{T}(x) \in \mathbb{R}^{h \times w}$, typically $h = 224, w = 224$. Note that channel dimensions are dropped for simplicity. The teacher receives the fully visible view $v_2$, whereas the student is given $v_1$ after randomly masking a subset of its patches, denoted as $\hat{v}_1$. The projected probability vectors are

$$\mathbf{a}_{s,v_1}^{[\text{cls}]} = \text{softmax}\left(\frac{\mathcal{P}_s^{[\text{cls}]}\left(\mathcal{E}_s^{[\text{cls}]}(\hat{v}_1)\right)}{\tau_s}\right), \quad \mathbf{a}_{t,v2}^{[\text{cls}]} = \text{softmax}\left(\frac{\mathcal{P}_t^{[\text{cls}]}\left(\mathcal{E}_t^{[\text{cls}]}(v_2)\right) - \mathbf{c}^{[\text{cls}]}}{\tau_t}\right), \quad (1)$$

and analogously $\mathbf{a}_s^{[\text{patch}]}, \mathbf{a}_t^{[\text{patch}]}$ for patch tokens are computed. Here $\tau_s$ and $\tau_t$ are temperature parameters and $\mathbf{c}^{[\text{cls}]}$ is a running mean (centering) of the cls features, which prevents collapse.

The student is updated by minimising the cross entropy between its predictions and those of the teacher. The cls token features are matched across views and the patch token features are matched *within* the same view.

$$\mathcal{L}_{\text{cls}} = -a_{t,v_2}^{[\text{cls}]} \log a_{s,v_1}^{[\text{cls}]} - a_{t,v_1}^{[\text{cls}]} \log a_{s,v_2}^{[\text{cls}]}, \quad \mathcal{L}_{\text{patch}} = -a_{t,v_1}^{[\text{patch}]} \log a_{s,v_1}^{[\text{patch}]} - a_{t,v_2}^{[\text{patch}]} \log a_{s,v_2}^{[\text{patch}]} \quad (2)$$

The overall iBoT objective is the weighted sum: $\mathcal{L}_{\text{iBot}} = \alpha_1 \times \mathcal{L}_{[\text{cls}]} + \alpha_2 \times \mathcal{L}_{[\text{patch}]}$, where $\alpha_1, \alpha_2 > 0$ balance the class- and patch-level signals. The teacher parameters are updated via an exponential moving average (EMA) of the student: $\theta_t \leftarrow \lambda\theta_t + (1 - \lambda)\theta_s$, with $\lambda$ following a cosine schedule from 0.996 to 1 during training [15].

## 3.2 Phase 2: Discovering Visual Concepts

While the global representations learned in Phase 1 capture overall scene identity, many downstream tasks demand finer, object-level understanding. Cognitive research shows that humans naturally organise visual input into coherent groups based on principles such as similarity, proximity, and continuity, a process known as perceptual grouping or segmentation [43, 44]. Inspired by this, we propose to decompose each image into $N$ distinct regions. Each region is represented by a dedicated concept token, as illustrated in Figure 2, which serves as a compact representation of that visual component. Several elements of this phase are informed by insights from the object-centric learning literature [45, 46] and are designed to promote the emergence of structured representations.

Given an augmented view $v$, the student and teacher ViT encoders produce patch-level features:

$$\mathbf{F}_s = \mathcal{E}_s^{[\text{patch}]}(\hat{v}), \qquad \mathbf{F}_t = \mathcal{E}_t^{[\text{patch}]}(v) \in \mathbb{R}^{S \times D} \qquad (3)$$

Here $S = S_x \times S_y$ denotes the total number of image patches, where $S_x = {}^h/p$ and $S_y = {}^w/p$ correspond to the number of patches along the height and width dimensions, respectively, given a patch stride of $p$. $D$ represents the embedding dimension of the ViT backbone.

A learnable matrix $\mathbf{C}^0 \in \mathbb{R}^{N \times D}$ is used to initialise $N$ concept queries, each intended to capture a distinct, high-level visual component. These concept tokens interact with the student patch features $\mathbf{F}_s$ via a lightweight clustering module $\mathcal{C}$, composed of $L$ Transformer decoder blocks. Within each block, cross-attention mechanisms enable the concept tokens to attend to and integrate information from the patch-level features. The cross-attention update at the $\ell$-th layer is given by:

$$\mathbf{Z}^{\ell+1} = \text{MHA}(\mathbf{Q} = \mathbf{Z}^\ell, \ \mathbf{K} = \mathbf{F}_s, \ \mathbf{V} = \mathbf{F}_s) + \mathbf{Z}^\ell, \qquad \ell = 0, \ldots, L - 1, \quad \mathbf{Z}^0 = \mathbf{C}^0,$$

where MHA denotes multi-head attention. This iterative process produces the final set of concept representations $\mathbf{Z} = \mathbf{Z}^L \in \mathbb{R}^{N \times D}$. Each decoder block also includes LayerNorm and feed-forward sublayers, though these components are omitted from the equation for brevity.

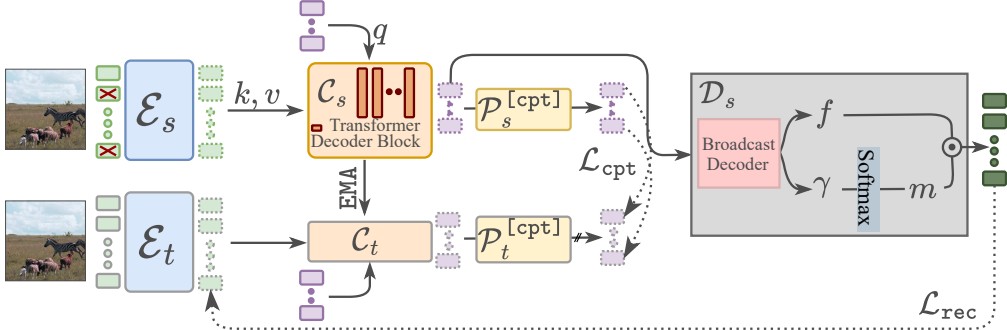

Figure 2: Overview of Phase 2. Given patch-level features from a masked student view, a set of learnable concept tokens iteratively interact with the patch grid via a lightweight clustering module to produce coherent region representations. The training is guided by two complementary objectives: masked concept distillation, enforced by the loss $\mathcal{L}_{\texttt{cpt}}$, and feature-space reconstruction, guided by $\mathcal{L}_{\texttt{rec}}$, both encouraging the emergence of meaningful and consistent region groupings.

Two design choices require clarification. First, stacking multiple cross-attention layers, $L$, allows each token to iteratively refine its region representation, querying patches based on increasingly informed prototypes. This results in sharper, more coherent region boundaries. Second, although many works apply entropy-based regularisation (e.g., Sinkhorn) to prevent token collapse, we observed no such issue with the concept tokens, thanks to Phase 1's spatially diverse features. We show in the ablations that adding Sinkhorn modestly reducing performance, so we omit it for simplicity.

To ensure the concept tokens are both meaningful and useful, we introduce two auxiliary objectives: concept-level distillation and feature-space reconstruction.

**Concept-level Distillation.** The first objective builds directly on the asymmetric masking strategy employed in Phase 1 and leverages it to guide concept-level learning. Given that the student network receives a *masked view* of the image, while the teacher processes the full, unmasked version. As a result, the student is encouraged to infer the structure of missing regions based on limited input, while aligning with the teacher's holistic understanding. The concept token sets from the student and teacher, denoted as $\mathbf{Z}_{s,v}$ and $\mathbf{Z}_{t,v}$ respectively, are projected through a projection head and compared using cross-entropy loss:

$$\mathbf{a}_{s,v}^{[\texttt{cpt}]} = \mathrm{softmax}\left(\frac{\mathcal{P}_s^{[\texttt{cpt}]}(\mathbf{Z}_{s,v})}{\tau_s}\right), \qquad \mathbf{a}_{t,v}^{[\texttt{cpt}]} = \mathrm{softmax}\left(\frac{\mathcal{P}_t^{[\texttt{cpt}]}(\mathbf{Z}_{t,v}) - \mathbf{c}^{[\texttt{cpt}]}}{\tau_t}\right), \quad (4)$$

where $\mathcal{P}_s^{[\texttt{cpt}]}$ and $\mathcal{P}_t^{[\texttt{cpt}]}$ denote the student and teacher projection heads for the concept tokens and $\mathbf{c}^{[\texttt{cpt}]}$ is a running mean of concept features. The concept-level distillation loss is then defined as:

$$\mathcal{L}_{\texttt{cpt}} = -\mathbf{a}_{t,v_1}^{[\texttt{cpt}]} \log \mathbf{a}_{s,v_1}^{[\texttt{cpt}]} - \mathbf{a}_{t,v_2}^{[\texttt{cpt}]} \log \mathbf{a}_{s,v_2}^{[\texttt{cpt}]} \quad (5)$$

Unlike the patch alignment objective in Phase 1, this loss encourages each concept token to capture the global structure of its corresponding region, despite partial masking. As such, it serves as a high-level signal that promotes consistency and the emergence of coherent region representations.

**Feature-space Reconstruction.** The second objective ensures that each concept token captures sufficient information to reconstruct the teacher's patch-level features. To achieve this, we use a lightweight spatial broadcast decoder $\mathcal{D}$, adapted from [47]. Each concept token is first broadcasted over a 2D spatial grid of size $S_x \times S_y$ and then augmented with learned positional embeddings. This results in a tensor of shape $S_x \times S_y \times D$, which is processed by a shared CNN (with weights shared across all concepts) to produce per-concept outputs of shape $S_x \times S_y \times (D+1)$.

Each per-concept output is split into two components: (i) a reconstructed patch feature map $f^n \in \mathbb{R}^{S_x \times S_y \times D}$, and (ii) a spatial confidence map $\gamma^n \in \mathbb{R}^{S_x \times S_y}$, where $n \in \{1, \ldots N\}$ indexes the $n$-th concept. To integrate the reconstructions from all concepts, we perform a softmax over the $N$ confidence maps $\gamma^1, \ldots, \gamma^N$ at each spatial location. This yields normalised attention weights $m^n$ for each concept:

$$m^n = \mathrm{softmax}\left(\frac{\gamma^n}{\kappa}\right) \in \mathbb{R}^{S_x \times S_y}, \quad \text{for } n = 1, \ldots, N, \quad (6)$$

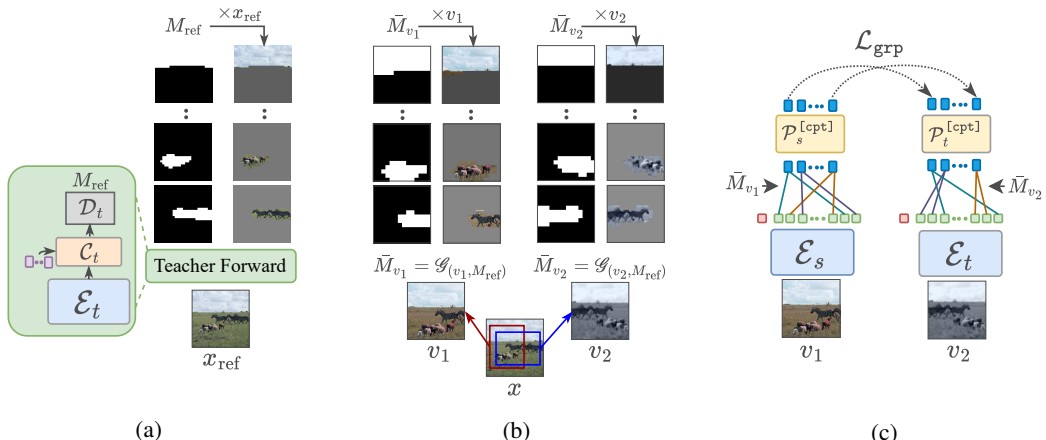

(a)                              (b)                              (c)

Figure 3: Phase 3 Overview. Given a reference image $x_{ref}$, the model generates $N$ attention masks $M_{\text{ref}}$, each capturing a learned concept (left). These masks are geometrically projected onto two augmented views $v_1$ and $v_2$, producing $\bar{M}_{v_1}$ and $\bar{M}_{v_2}$ (middle). Patch tokens are then aggregated per concept using the projected masks, yielding $N$ concept vectors. These are passed through a projection head and aligned via cross-entropy loss to enforce concept-level consistency across views (right). Masks are overlaid on images for visualisation only.

where the softmax is computed independently at each spatial location across the $N$ concepts, and $\kappa$ is a temperature parameter controlling the sharpness of the distribution.

The final reconstructed feature map is obtained by computing a weighted sum of the per-concept reconstructions: $\widehat{\mathbf{F}} = \sum_{n=1}^{N} m^n \odot f^n \in \mathbb{R}^{S_x \times S_y \times D}$, where $\odot$ denotes element-wise multiplication broadcasted over the feature dimension $D$. Finally, the reconstruction loss is defined as:

$$\mathcal{L}_{\text{rec}} = \left\| \widehat{\mathbf{F}}_{s,v} - \mathbf{F}_{t,v} \right\|^2 \qquad (7)$$

This objective encourages the concept tokens to distribute their representational capacity over different spatial regions, ensuring a non-redundant and semantically rich decomposition of the scene.

The overall loss for Phase 2 combines both objectives: $\mathcal{L}_{\text{Phase2}} = \alpha_3 \mathcal{L}_{\text{cpt}} + \alpha_4 \mathcal{L}_{\text{rec}}$, with weights $\alpha_3, \alpha_4 > 0$. These terms are optimised jointly with Phase 1.

### 3.3 Phase 3: View-Invariant Alignment of Discovered Concepts

Once humans form abstract representations of concepts in a scene, they are able to recognise these same entities under different lighting conditions, perspectives, or spatial configurations. This capacity, often referred to as view-invariant recognition, allows us to perceive an object as the same despite visual variations that arise naturally in everyday environments [48, 49].

Following the discovery of visual concepts in Phase 2, we aim to equip our model with a similar ability by maintaining consistent concept representations across diverse augmented views. While Phase 2 clusters spatial features into a fixed number of tokens, there is no guarantee that the same concept appears in every augmented view, nor that it will be positioned or ordered consistently. Therefore, trying to match across views, especially under strong augmentations, can be unreliable. To address this, we design a guided alignment mechanism that uses a stable *reference image* as an anchor for transferring concept regions to multiple views via geometric transformations.

Given an image $x$, we generate a lightly augmented reference view with no cropping, $x_{\text{ref}}$. The teacher processes $x_{\text{ref}}$ to produce $m_{\text{ref}}$, which is then thresholded to obtain binary masks $\mathbf{M}_{\text{ref}} \in \{0,1\}^{S_{\text{ref}} \times N}$, where each mask indicates the spatial support of a concept.

Using crop metadata, we project the $\mathbf{M}_{\text{ref}}$ into the coordinate grids of $v_1, v_2$, obtaining $\bar{\mathbf{M}}_{v_1}, \bar{\mathbf{M}}_{v_2} \in \{0,1\}^{S \times N}$. We then perform masked feature aggregation to obtain pooled concept features:

$$\bar{\mathbf{z}}_{s,v_k}^n = \frac{\sum_{i,j} \bar{M}_{v_k}^{i,j,n} \cdot \mathbf{F}_{s,v_k}^{i,j}}{\sum_{i,j} \bar{M}_{v_k}^{i,j,n} + \varepsilon}, \quad k \in \{1,2\} \qquad (8)$$

These per-concept vectors are projected via a shared head $\mathcal{P}^{[\text{cpt}]}$ and compared across views using cross-entropy loss:

$$\mathcal{L}_{\text{grp}} = -\sum_{n \in \mathcal{V}} \left[ \bar{\mathbf{a}}_{s,v_1}^{[n]} \log \bar{\mathbf{a}}_{s,v_2}^{[n]} + \bar{\mathbf{a}}_{s,v_2}^{[n]} \log \bar{\mathbf{a}}_{s,v_1}^{[n]} \right] \tag{9}$$

where $\mathcal{V}$ includes concept indices visible in both views. To ensure stability, we delay applying $\mathcal{L}_{\text{grp}}$ until concept tokens from Phase 2 have sufficiently converged. Full derivations, including mask transfer, visibility conditions, and alignment schedules, are provided in the Appendix.

The overall loss in Phase 3 combines all preceding objectives: $\mathcal{L}_{\text{CG-SSL}} = \alpha_1 \mathcal{L}_{\text{cls}} + \alpha_2 \mathcal{L}_{\text{patch}} + \alpha_3 \mathcal{L}_{\text{cpt}} + \alpha_4 \mathcal{L}_{\text{rec}} + \alpha_5 \mathcal{L}_{\text{grp}}$. For simplicity, we set $\alpha_i = 1$ for $i = 1, \ldots, 5$.

| Model | ADE-20K | | Pascal-VOC | | Cityscapes | |
|---|---|---|---|---|---|---|
| | $k$-NN | Linear | $k$-NN | Linear | $k$-NN | Linear |
| ViT-Small | | | | | | |
| SiT | 14.9 | 20.6 | 35.8 | 45.4 | – | – |
| MC-SSL | 18.2 | 22.2 | 54.5 | 61.4 | – | – |
| Dino | 15.5 | 18.5 | 36.3 | 41.6 | – | – |
| iBot | 19.3 | 23.6 | 48.1 | 56.4 | – | – |
| CG-SSL | **25.8** | **28.9** | **60.8** | **66.7** | **34.2** | **39.1** |
| CAPI (LVD-142M) | 20.6 | 25.8 | 50.5 | 59.4 | 31.3 | 36.5 |
| ViT-Base | | | | | | |
| MAE | 17.3 | 23.1 | 46.8 | 55.6 | – | – |
| Data2Vec 2.0 | 16.8 | 21.2 | 47.4 | 48.9 | – | – |
| MC-SSL | 23.3 | 27.2 | 60.8 | 65.1 | – | – |
| iBot | 22.9 | 27.3 | 57.1 | 65.4 | – | – |
| CG-SSL | **28.3** | **31.5** | **62.7** | **69.0** | 35.2 | 40.0 |
| CAPI (LVD-142M) | 28.3 | 33.6 | 61.8 | 70.3 | 36.9 | 41.9 |
| ViT-Large | | | | | | |
| MAE | 21.5 | 27.4 | 53.7 | 61.5 | 32.8 | 38.5 |
| Data2Vec 2.0 | 24.2 | 27.6 | 57.5 | 58.1 | 32.8 | 38.2 |
| iBot | 26.0 | 30.7 | 60.2 | 68.8 | 35.7 | 39.8 |
| CAPI | 29.2 | 34.4 | 60.7 | 69.7 | 35.6 | 41.7 |
| CG-SSL | **30.5** | **35.6** | **64.0** | **70.6** | **36.3** | **45.5** |
| DINOv2 (LVD-142M) | 34.0 | 39.0 | 63.0 | 72.8 | 42.0 | 46.8 |
| CAPI (LVD-142M) | 32.1 | 37.2 | 63.8 | 72.7 | 38.9 | 44.3 |

Table 1: Comparison with state-of-the-art methods on semantic segmentation using frozen features. We report $k$-NN and linear probe performance. CG-SSL consistently outperforms prior approaches, highlighting the quality of its learned representations.

# 4 Experiments

We evaluate the effectiveness of CG-SSL framework through a series of experiments. We begin by outlining our experimental setup, including datasets, architectures, and implementation details. Next, we report quantitative and qualitative results on tasks involving whole-image understanding and dense prediction. Finally, we conduct ablation studies to validate key components of our method.

## 4.1 Experimental Setup

**Pretraining Dataset:** Our main models are pretrained on ImageNet-1K without labels. For ablation studies, we pretrain on a combination of dense datasets, namely PASCAL VOC [50], Visual Genome [51], and MS-COCO [52], which together provide approximately 170K diverse images.

**Implementation Details:** We use ViT backbone with patch size $16 \times 16$, following standard ViT-S/16, ViT-B/16, and ViT-L/16. We use $N = 4$ concept tokens and the clustering module $\mathcal{C}$ consists of $L = 4$ transformer decoder blocks. The output of $\mathcal{C}$, along with the encoder's [CLS] and [patch] tokens, are passed through a shared projection head comprising two linear layers with 2048 units and GELU activations, followed by a 256-dimensional bottleneck. The output is L2-normalised and projected into an 8192-dimensional embedding space. All models are trained using the AdamW optimiser with a cosine learning rate schedule and an effective batch size of 256 distributed across 8 GPUs. We train ViT-S for 800 epochs, ViT-B for 500 epochs, and ViT-L for 300 epochs. Further training and ablation details are included in the Appendix.

| | ViT-S/16 | | ViT-B/16 | |
|---|---|---|---|---|
| | KNN | Linear | KNN | Linear |
| iBot | 75.2 | 77.9 | 77.1 | 79.5 |
| CG-SSL | 74.7 | 77.7 | 76.8 | 79.0 |
| CAPI(LVD-142M) | – | 71.5 | – | 79.6 |

(a) Multi-class Linear Classification

| | ViT-S/16 | | | ViT-B/16 | | |
|---|---|---|---|---|---|---|
| | Pascal | VGenome | COCO | Pascal | VGenome | COCO |
| iBot | 89.4 | 30.3 | 57.4 | 90.1 | 30.4 | 58.2 |
| CG-SSL | **90.3** | **31.3** | **58.3** | **91.1** | **31.7** | **58.8** |

(b) Multi-label Linear Classification

Table 2: Image classification results employing ViT-S/16 and ViT-B/16 backbones.

## 4.2 Results

**Dense Image Understanding.** A core motivation behind CG-SSL is to produce semantically rich and spatially localised representations via patch-level features. To evaluate the quality of these representations, we conduct experiments on dense prediction tasks, specifically semantic segmentation. We follow the $k$-NN and linear probe evaluation protocols from [53] and report mIoU on three standard benchmarks: ADE20K [54], PASCAL VOC [55], and Cityscapes [56], refer to Table 1.

Notably, CG-SSL achieves high $k$-NN performance across all datasets, often matching or even exceeding the performance of strong baselines under linear probing. This is particularly impressive, as $k$-NN relies purely on raw feature similarity without any additional training, making it a direct measure of the intrinsic quality of the learned representations. For example, on Pascal VOC, CG-SSL outperforms prior SOTA methods pre-trained on $100\times$ more data, i.e. CAPI, by a wide margin of over 10 mIoU in the $k$-NN setting for the ViT-S backbone. This substantial boost under a non-parametric evaluation setup demonstrates that CG-SSL learns highly discriminative and well-structured representations suitable for spatial reasoning and region-based understanding.

**Image Classification.** We evaluate CG-SSL using a standard linear probing setup on multi-class and multi-label classification tasks. As shown in Table 2, while CG-SSL shows slightly lower performance than iBOT on ImageNet (the pretraining dataset), it consistently outperforms iBOT on other datasets such as Pascal, COCO, and Visual Genome. This is expected as our method balances both global representation and dense localisation, which may slightly reduce image-level discrimination on seen data but yields stronger generalisation to unseen domains.

**Qualitative Comparison of Dense Feature Representations via PCA.** We adopt the qualitative feature analysis proposed in CAPI [53], applying PCA to the dense output features. As shown in Figure 4, we visualise the first three principal components as an RGB composite and compare CG-SSL against SOTA vision models employing ViT-L backbone, except for I-JEPA, which uses ViT-H. Among all methods, CG-SSL produces some of the most discriminative and spatially coherent feature maps. The visualisations reveal a clear emergence of object boundaries with minimal noise in uniform regions. Compared to CAPI and DINOv2, CG-SSL features appear less noisy. Moreover, in contrast to masked image modeling (MIM) methods, CG-SSL focuses more effectively on semantically meaningful regions of the image. More visualisations are in Appendix.

| **Image** | **MAE** | **I-Jepa** | **D2V2** | **DI-NOv2+r** | **CAPI** | **CG-SSL** |
|---|---|---|---|---|---|---|

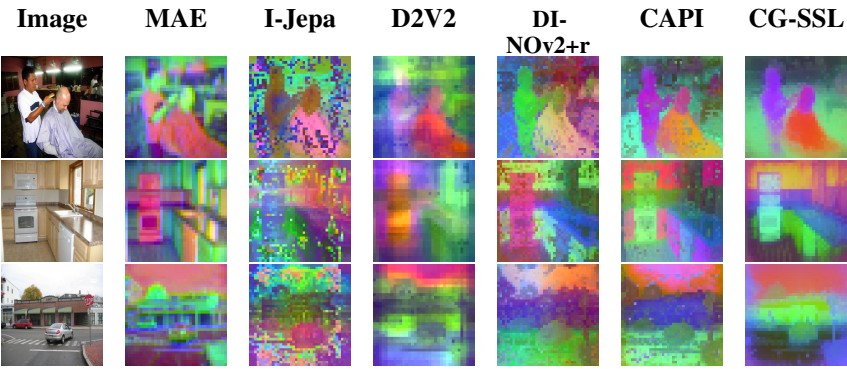

Figure 4: Comparison of feature visualisations produced by CG-SSL ViT-L/16 and SOTA methods on 560-pixel resolution images. We apply PCA to the dense feature maps. The first column displays a composite RGB image formed from the first three principal components.

As further qualitative assesment, we visualise the attention maps produced by our model on randomly selected images outside the ImageNet-1K dataset (Figure 5). Each map highlights a distinct concept,

showing the model's ability to localise semantically meaningful regions, even under occlusion. For instance, in the second image, the model accurately identifies the visible part of the cow despite being partially obscured by a person.

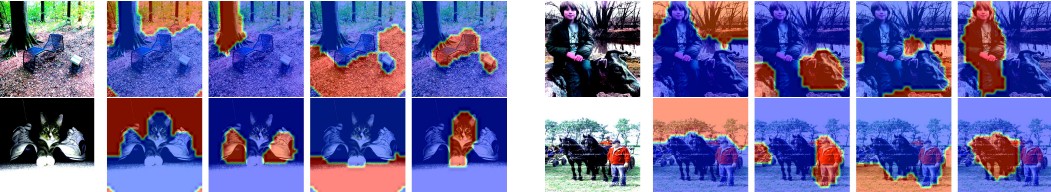

Figure 5: Attention maps for learned concepts using ViT-L/16 pretrained on ImageNet-1K.

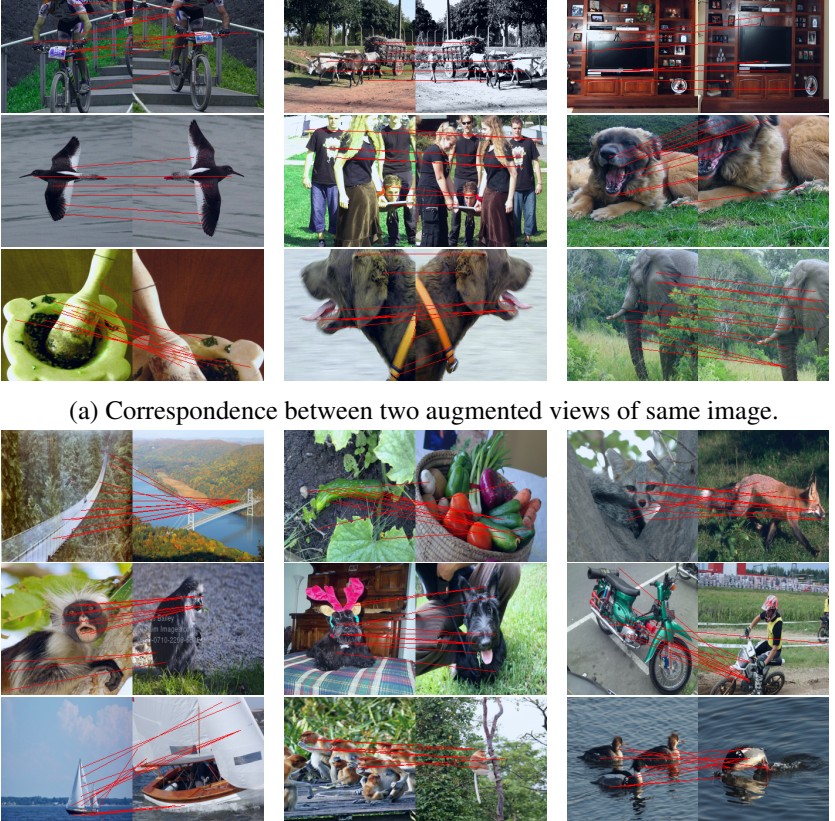

(a) Correspondence between two augmented views of same image.

(a) Correspondence between two different images from same class.

Figure 6: Top 12 patch correspondences extracted by CG-SSL with ViT-L/16.

**Sparse Correspondence.** We tackle a sparse correspondence task in which patches from two images belonging to the same semantic class are expected to be matched. To assess performance, we visualise the top 12 correspondences with the highest self-attention scores, obtained from a ViT-L/16 model pre-trained on ImageNet-1K using our CG-SSL framework. The image pairs are sampled from the ImageNet validation set.

Figure 6 presents representative examples of such correspondences. In Figure 6a, CG-SSL exhibits near-perfect correspondence matching between two augmented views of the same image, accurately aligning nearly all patch pairs.

Moreover, as depicted in Figure 6b, CG-SSL effectively establishes meaningful correspondences across two different images of the same class, despite substantial differences in texture, color, pose,

and background context. These results underscore the robustness and generalisation capabilities of the learned representations, demonstrating their suitability for fine-grained, patch-level retrieval tasks.

**(a)**

|  | ADE | Pascal |
|---|---|---|
| DINO | 15.6 | 35.7 |
| iBot | 18.8 | 47.2 |
| CG-SSL [Stage 1,2] | 21.4 | 53.4 |
| CG-SSL [Stage 1,2,3] | **24.3** | **56.1** |

**(b)**

|  | ADE | Pascal |
|---|---|---|
| **Frozen** | | |
| Kmeans | 23.1 | 54.8 |
| Ours [Frozen] | 22.9 | 54.6 |
| **Learnable** | | |
| SlotAttn | 19.8 | 52.2 |
| Ours | **24.3** | **56.1** |

**(c)**

|  | ADE | Pascal |
|---|---|---|
| 2 | 23.8 | 55.3 |
| 4 | 24.1 | 55.9 |
| 6 | **24.3** | **56.1** |
| 8 | 24.0 | 55.7 |
| 12 | 23.8 | 55.5 |

**(d)**

|  | ADE | Pascal |
|---|---|---|
| 1 | 23.9 | 55.9 |
| 2 | 24.0 | 56.0 |
| 4 | **24.3** | **56.1** |
| 8 | 23.8 | 55.9 |

**(e)**

| $\kappa$ | ADE | Pascal |
|---|---|---|
| 0.1 | 23.8 | 55.9 |
| 0.3 | 24.1 | 56.0 |
| 0.5 | **24.3** | **56.1** |
| 0.8 | 23.7 | 55.1 |
| 1 | 23.3 | 55.0 |

**(f)**

|  | ADE | Pascal |
|---|---|---|
| Pixels | 23.8 | 55.6 |
| Features | **24.3** | **56.1** |

**(g)**

| Shared | ADE | Pascal |
|---|---|---|
| ✓ | **24.3** | **56.1** |
| ✗ | 23.4 | 49.8 |

**(h)**

| $\mathcal{L}_{\texttt{patch}}$ | ADE | Pascal |
|---|---|---|
| ✓ | **24.3** | **56.1** |
| ✗ | 24.1 | 56.0 |

Table 3: Ablation study of the design choices in our framework. We report linear performance on image segmentation tasks. We highlight the default setting in gray, and bold the best-performing solution. An in-depth analysis of these results is provided in the Appendix.

## 4.3 Ablation Studies

We conduct a comprehensive set of ablations to show the impact of key design choices in CG-SSL, including (a) incremental component build-up across training phases, (b) learnable vs. frozen clustering modules, (c) varying the number of concept tokens, (d) decoder depth in the clustering module, (e) reconstruction temperature, (f) choice of reconstruction target, (g) projection head sharing between [CLS] and concept tokens, and (h) retaining patch-level loss in later phases. Results are reported in Tables 3a-3h. For detailed analysis and additional ablations, see Appendix.

## 5 Discussion and Concluding Remarks

We introduced **CG-SSL**, a structured self-supervised framework that discovers, organises, and aligns semantic concepts across views without requiring manual supervision or pretrained models. Through a three-phase curriculum, global representation learning, concept discovery, and view-invariant alignment, CG-SSL bridges the gap between global discriminative features and spatially coherent representations. Our experiments shows that CG-SSL achieves strong performance on both classification and segmentation tasks, outperforming several baselines and even rivaling models trained on substantially larger datasets. Notably, the use of concept tokens enables interpretability and spatial localisation that are difficult to achieve with conventional invariance-based methods.

**Limitations and Future Work.** While CG-SSL provides structured and interpretable representations, the number of concept tokens is fixed across images, which may be suboptimal for scenes with varying complexity. Looking forward, we aim to explore adaptive mechanisms for determining the number of concept tokens per image and to extend CG-SSL to video, enabling temporal concept tracking. Importantly, this work was developed by a group with modest computational resources. As was the case with DINOv2, which achieved significant performance gains by scaling and optimising iBOT, we believe CG-SSL has similar potential to benefit from scaling and extensive hyperparameter tuning. We hope the community can build upon this foundation and explore its full capacity.

## 6 Acknowledgment

This research was supported by the UKRI AHRC CoSTAR National Lab for Creative Industries R&D as part of the Creative AI Futures

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

# Technical Appendices and Supplementary Material

## A   Additional Details For Phase 3: View-Invariant Alignment

Once the model learns to decompose an image into coherent concepts (Phase 2), the next step is to ensure that these concepts are consistently recognised across different augmented views. Human vision naturally achieves this through view-invariant recognition, maintaining stable representations of entities under transformations such as cropping, flipping, or color variation [48, 49].

However, directly matching concept tokens across views is unreliable due to their permutation-invariant and view-specific nature. To address this, we design a guided alignment mechanism that uses a stable *reference image* as an anchor for transferring concept regions to multiple views via geometric transformations.

For each input image $x \in \mathbb{R}^{H \times W}$, we generate:

- A *reference view* $x_{\text{ref}} \in \mathbb{R}^{\hat{h} \times \hat{w}}$ via light augmentations (no cropping), fed unmasked to the teacher.
- Two strongly augmented views $v_1, v_2 \sim \mathcal{T}(x) \in \mathbb{R}^{h \times w}$, each associated with crop parameters:

$$\phi^{(k)} = \left( x_0^{(k)}, y_0^{(k)}, w_{\text{crop}}^{(k)}, h_{\text{crop}}^{(k)}, \text{flip}^{(k)} \right), \quad k \in \{1, 2\} \tag{10}$$

The teacher processes $x_{\text{ref}}$ to produce concept attention masks $\mathbf{M}_{\text{ref}} \in \{0, 1\}^{S_{\text{ref}} \times N}$, where $S_{\text{ref}} = S_{\text{ref},x} \times S_{\text{ref},y}$ is the number of patches in the reference view and $N$ is the number of concept tokens.

**Mask Transfer via Geometry.** Let $(i, j)$ denote a patch coordinate in the reference grid. We define its normalized center as:

$$u = \frac{i + 0.5}{S_{\text{ref},x}}, \quad v = \frac{j + 0.5}{S_{\text{ref},y}} \tag{11}$$

This patch is visible in view $v_k$ if:

$$x_0^{*(k)} \le u < x_0^{*(k)} + w_{\text{crop}}^{*(k)}, \quad y_0^{*(k)} \le v < y_0^{*(k)} + h_{\text{crop}}^{*(k)} \tag{12}$$

where the crop parameters are normalized by the original image size $H \times W$. For visible patches, we compute the projected patch index $(i', j')$ in view $v_k$:

$$i' = \left\lfloor S_{v_k,x} \cdot \frac{u - x_0^{*(k)}}{w_{\text{crop}}^{*(k)}} \right\rfloor, \quad j' = \left\lfloor S_{v_k,y} \cdot \frac{v - y_0^{*(k)}}{h_{\text{crop}}^{*(k)}} \right\rfloor \tag{13}$$

If horizontal flipping is applied ($\text{flip}^{(k)} = 1$), then $i' \leftarrow S_{v_k,x} - 1 - i'$. The transferred binary mask $\bar{\mathbf{M}}_{v_k}^n \in \{0, 1\}^{S_{v_k}}$ for concept $n$ is defined as:

$$\bar{M}_{v_k}^{n,i',j'} = \begin{cases} M_{\text{ref}}^{n,i,j}, & \text{if Eq. 12 holds}, \\ 0, & \text{otherwise} \end{cases} \tag{14}$$

**Masked Feature Aggregation.** Let $\mathbf{F}_{s,v_k} \in \mathbb{R}^{S_{v_k} \times D}$ denote the student's patch features for view $v_k$. We compute aggregated concept features by masked average pooling over patches:

$$\mathbf{z}_{s,v_k}^n = \frac{\sum_{i',j'} \bar{M}_{v_k}^{n,i',j'} \cdot \mathbf{f}_{s,v_k}^{i',j'}}{\sum_{i',j'} \bar{M}_{v_k}^{n,i',j'} + \varepsilon} \tag{15}$$

Only concepts that are visible in both $v_1$ and $v_2$ (i.e., with non-zero denominator in both) are aligned.

**Concept-Level Alignment Loss.** Each aggregated concept feature $\mathbf{z}_{s,v_k}^n$ is passed through a shared projection head $\mathcal{P}^{[\text{cpt}]}$, and softmax-normalized:

$$\mathbf{a}_{s,v_k}^{[n]} = \text{softmax}\left(\frac{\mathcal{P}^{[\text{cpt}]}(\mathbf{z}_{s,v_k}^n)}{\tau_s}\right) \tag{16}$$

The group-level consistency loss is defined as cross-entropy across matching concepts:

$$\mathcal{L}_{\text{grp}} = -\sum_{n \in \mathcal{V}}\left[\mathbf{a}_{s,v_2}^{[n]} \cdot \log \mathbf{a}_{s,v_1}^{[n]} + \mathbf{a}_{s,v_1}^{[n]} \cdot \log \mathbf{a}_{s,v_2}^{[n]}\right] \tag{17}$$

where $\mathcal{V}$ is the set of concept indices visible in both views. This loss enforces consistent representations for matched concepts under geometric variation.

**Training Schedule.** To ensure stable alignment, we delay the application of $\mathcal{L}_{\text{grp}}$ until the model has learned coherent concept masks from Phase 2.

$$\mathcal{L}_{\text{total}} = \alpha_1 \mathcal{L}_{[\text{cls}]} + \alpha_2 \mathcal{L}_{[\text{patch}]} + \alpha_3 \mathcal{L}_{\text{cpt}} + \alpha_4 \mathcal{L}_{\text{rec}} + \alpha_5 \mathcal{L}_{\text{grp}} \tag{18}$$

# B  Experimental Setup

**Implementation Details.** We use ViT backbone with patch size $16 \times 16$, following standard ViT-S/16, ViT-B/16, and ViT-L/16 configurations with embedding dimensions of 384, 768, and 1024, respectively. Each encoder outputs a sequence of patch tokens and a global `[CLS]` token.

For the clustering module $\mathcal{C}$, we use $N = 4$ concept tokens and $\mathcal{C}$ consists of $L = 4$ transformer decoder blocks. The output of $\mathcal{C}$, along with the encoder's `[CLS]` token and `[patch]` tokens, is passed through a shared projection head comprising two linear layers with 2048 units and GELU activations, followed by a 256-dimensional bottleneck. The output is L2-normalised and projected into an 8192-dimensional embedding space.

The broadcast decoder $\mathcal{D}$, adapted from [47], reconstructs teacher patch features from the concept tokens. Each token is broadcasted to the patch grid size, augmented with learned positional encodings, and processed with a shared MLP to produce patch-wise reconstructions $f^n$ and unnormalised attention weights $\gamma^n$; the final reconstruction is computed using softmax-normalised weights with temperature $\kappa = 0.5$. The decoder output dimension $D$ matches the backbone's embedding size.

For the reference image in Phase 3, we use a higher-resolution input of $448 \times 448$, resulting in a denser set of patch tokens, enhancing the quality of concept representations. Since there is no backpropagation from the teacher model, the increased resolution introduces minimal computational overhead.

All models are trained using the AdamW optimiser with a cosine learning rate schedule. We largely follow the iBOT pre-training hyperparameters, with the exception of increasing the masking ratio from 30% to 40%. We trained the models with effective batch size of 256 distributed across 8 GPUs on ViT-S for 800 epochs, ViT-B for 600 epochs, and ViT-L for 400 epochs.

For ViT-S/16, Phase 1 is applied for the first 50% of the training epochs, followed by Stages 1+2 for the next 10%, and the full pipeline (Stages 1+2+3) for the final 40%. These phase proportions are based on empirical intuition and have not been exhaustively tuned. For ViT-B/16 and ViT-L/16, we initialise from iBOT-pretrained weights to avoid redundant computation and reduce energy consumption. To ensure a fair comparison, we also continue training the original iBOT models under our schedule. However, we observe that their performance degrades with further training when using the original iBOT objectives.

**Time and Memory Requirements.** CG-SSL takes approximately 24 minutes per epoch to pre-train a ViT-B/16 model using 8 GPUs with an effective batch size of 256. Compared to iBOT, our method takes about 6.5 minutes longer per epoch and uses roughly 4.6 GB more GPU memory under the same training setup. While CG-SSL is approximately one epoch per hour slower, the added compute

is a reasonable trade-off for the improved spatial consistency, interpretability, and robustness of the learned representations.

## C   Visualisations

**Qualitative Comparison of Dense Feature Representations via PCA.** We adopt the qualitative feature analysis methodology proposed in CAPI [53], applying PCA to the dense output features, $\mathcal{E}_t^{\texttt{[patch]}}(x)$, extracted across all images. As shown in Figure 7, the second column visualises the first three principal components as an RGB composite, while the subsequent six columns show each of the first six components individually.

Each component highlights distinct semantic regions, revealing that CG-SSL encodes meaningful visual concepts. Notably, the components differentiate object parts from the background, indicating the model's ability to disentangle structured elements of the scene. This visualisation underscores the spatially localised and interpretable nature of CG-SSL's learned representations.

**Pattern Layout of CLS vs Concept Tokens.** For this experiment, we used the MSCOCO dataset, known for its visually rich and dense scenes. We extracted CLS token features and concept token features using a ViT-S/16 model pre-trained on ImageNet-1K. The clustering results, based on MSCOCO images, are shown in Figure 8 and Figure 9, respectively. By examining the top clusters, we observed several notable patterns.

CLS token features tend to capture the global context or overall "scene gist" of an image. This is evident in Figure 8, where images are grouped based on their broader visual setting. For instance, airplanes in the sky are clustered together, while those on the ground form a separate group. These clusters reflect how CLS tokens prioritise high-level scene understanding.

In contrast, concept token features focus more on localised and object-specific information. For visualisation clarity, we overlay the attention maps of each concept token on the images, highlighted in red. As shown in Figure 9, unlike CLS tokens, background elements are less influential in these features, leading to tighter focus on object-centric regions. For example, airplanes are clustered irrespective of whether they are flying or on the ground, and elephants are grouped across varying backgrounds. Even when only partial object visibility is present, concept tokens effectively capture the object identity.

A particularly interesting observation is the emergence of fine-grained clusters. In the last row of Figure 9, we see a distinct cluster composed entirely of hands from different images, activities, and viewpoints. This suggests that concept tokens are not only able to localise objects but also capture consistent semantic parts across varied visual contexts.

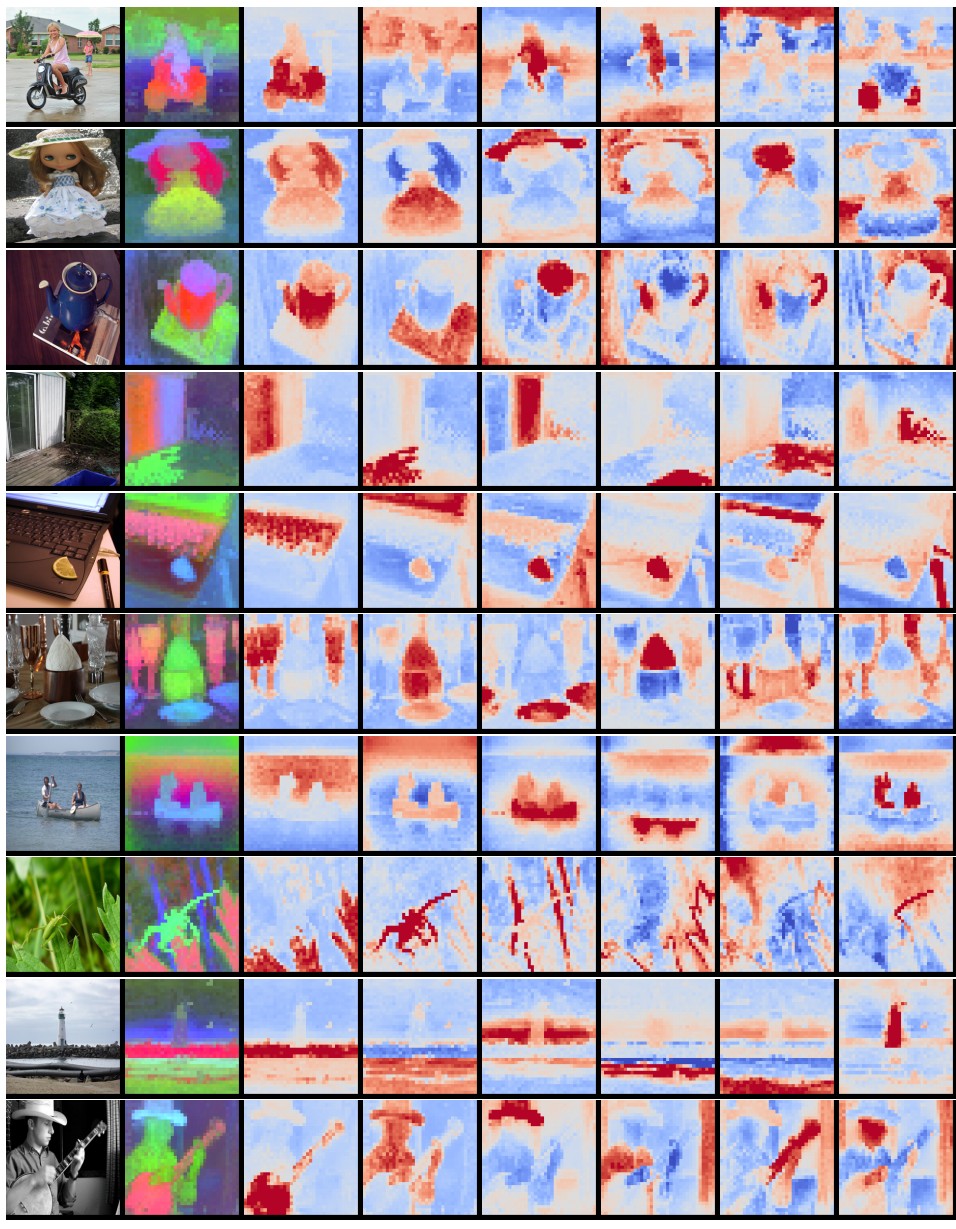

Figure 7: Visualisation of the features produced by CG-SSL ViT-L/16 applied to images at 560 pixel resolution. Images are randomely selected from validation set of ImageNet-1K dataset.

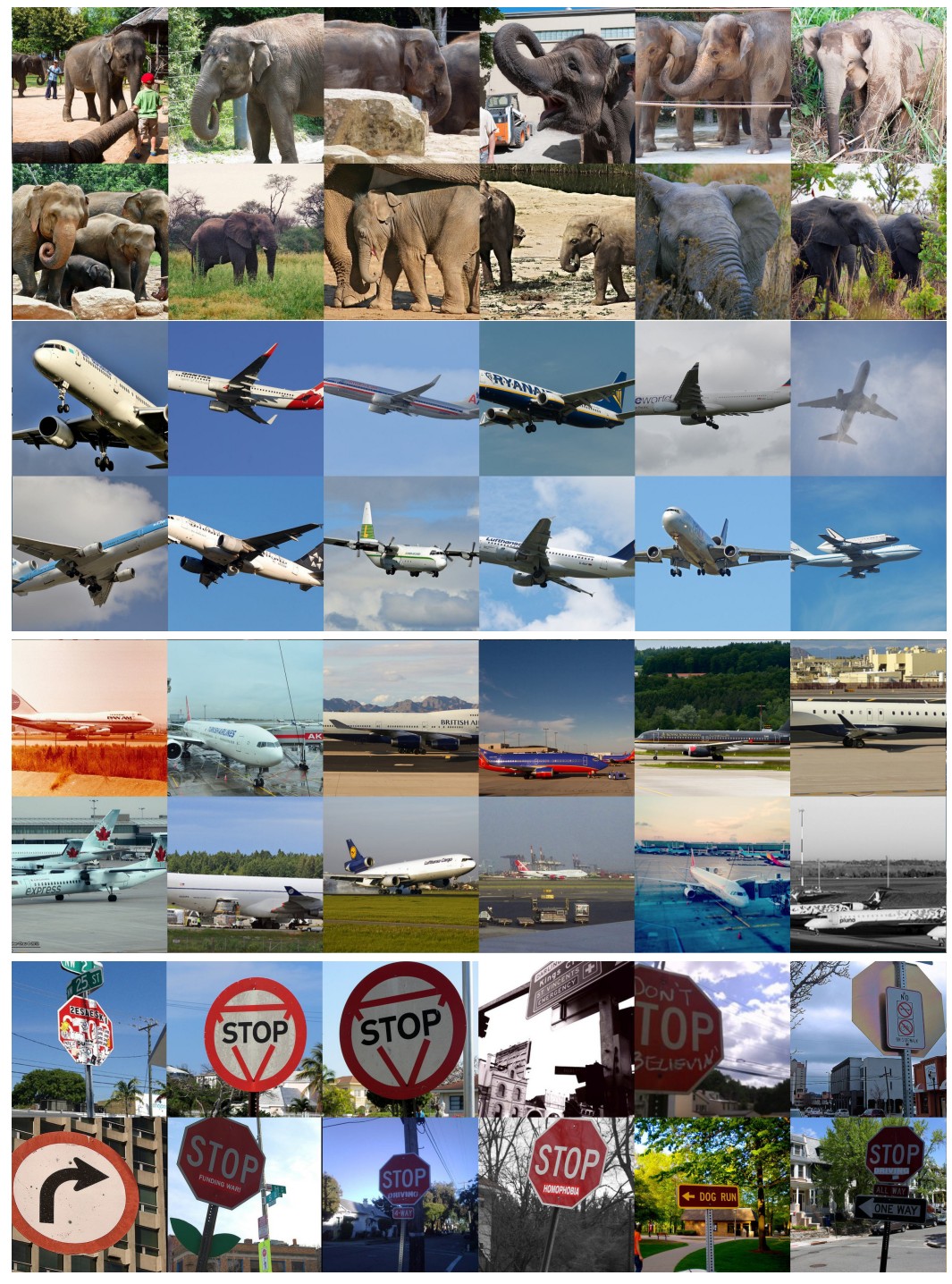

Figure 8: Visualisation for pattern layout of [CLS] token.

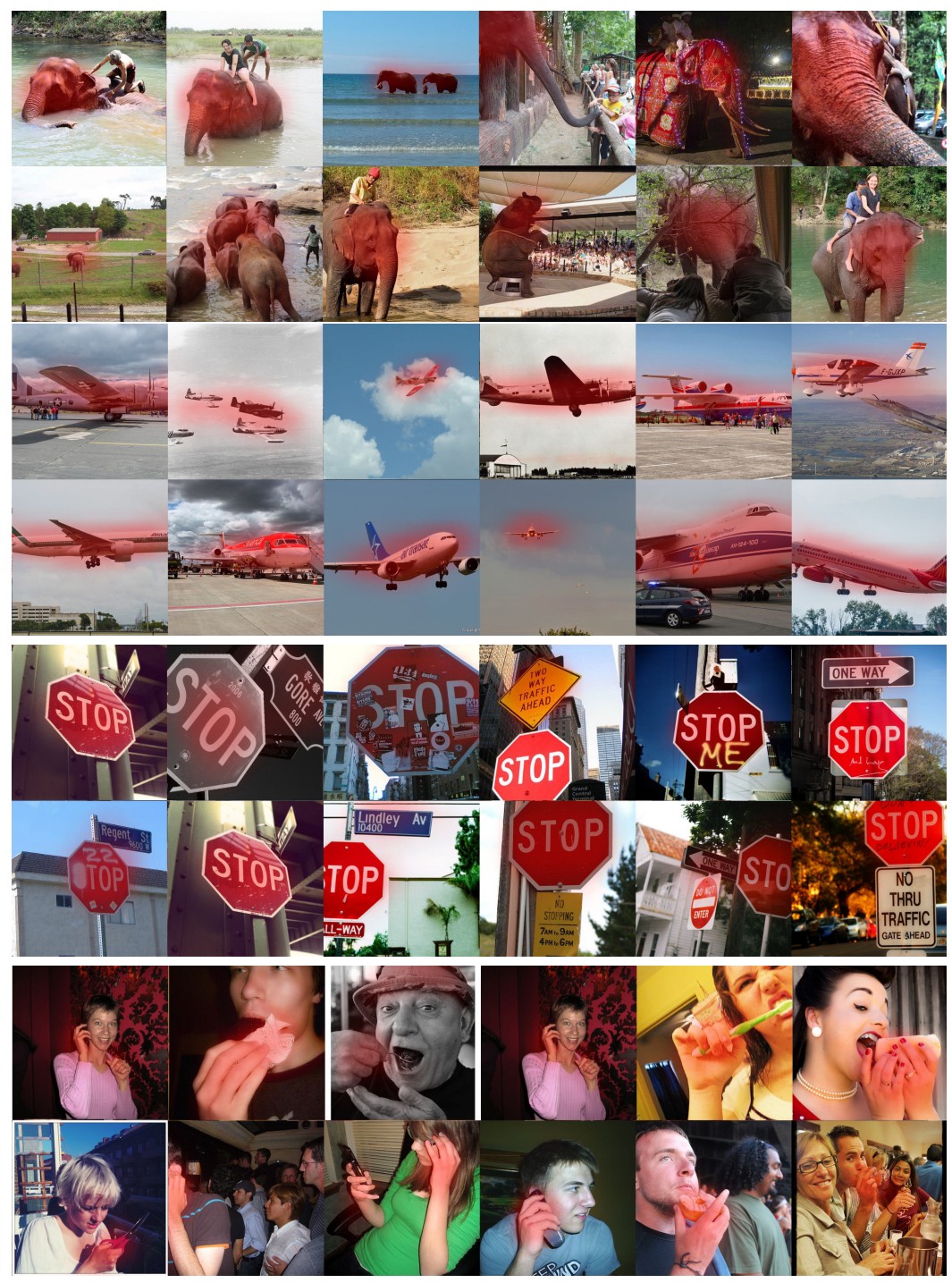

Figure 9: Visualisation for pattern layout of concept tokens.

# D    Detailed Ablation Analysis

We conduct comprehensive ablation studies to systematically analyse the influence of various design choices on our model's performance. Given resource constraints, we pretrain our model using a selected combination of dense datasets, namely PASCAL VOC, MS-COCO, and Visual Genome, which contains approximately 170K diverse samples. Our training employs the ViT-S/16 architecture for a total of 500 epochs. Although utilising 10% of ImageNet for ablation studies might seem a more conventional choice, we deliberately opt for these datasets due to their inherent complexity, characterised by crowded scenes and multiple objects, rather than the dominant, centrally located subjects typically found in ImageNet images. This selection enables us to rigorously evaluate our model's capability to extract meaningful representations from complex and cluttered visual scenes, an important aspect often underexplored in current self-supervised learning research. To ensure a fair comparison, we pretrain our baseline method, iBot, under identical conditions using the same dataset combination.

Our initial goal was to conduct the entire study exclusively on these challenging datasets. However, given the community's emphasis on ImageNet benchmarks, we include ImageNet-based comparisons to provide a clearer context for our findings. While our approach already outperforms state-of-the-art methods, we believe that conducting ablation studies directly on ImageNet could further enhance our results.

All numbers are mean intersection-over-union mIoU on ADE20K / Pascal-VOC using the ViT-S/16 backbone with a frozen linear probe, as in Table 3 of the main paper.

## D.1    Curriculum Build-up Across the Three Phases

As shown in Table 4, while DINO and iBOT offer strong global representations, they fall short on dense prediction tasks, reaching only 15.6/35.7 and 18.8/47.2 mIoU on ADE20K and Pascal, respectively. Adding Phase 2 improves the structure of learned features by grouping spatial regions into object-like concepts, which raises performance to 21.4/53.4. Phase 3 further aligns these concepts across views, enforcing spatial consistency and yielding the final boost to 24.3/56.1. This progression shows the importance of both discovering region-level structure and ensuring it transfers across viewpoints.

| Model / Phases | ADE | Pascal |
|---|---|---|
| DINO baseline | 15.6 | 35.7 |
| iBOT baseline | 18.8 | 47.2 |
| CG-SSL (Stage 1 + 2) | 21.4 | 53.4 |
| **CG-SSL (1 + 2 + 3)** | **24.3** | **56.1** |

Table 4: Effect of progressively adding the three curriculum phases.

## D.2    Clustering Strategy for Concept Tokens

We explored several alternatives to the learnable clustering module in Phase 2 as shown in Table 5. Surprisingly, even a simple $k$-means clustering on the reference image, followed by geometric alignment, achieved 23.1/54.8 on ADE20K/Pascal. This highlights the strength of Phase 3. However, this setup adds nontrivial computational overhead due to the per-image clustering step, making it less scalable.

When we used our clustering module but froze it after initial training, performance dropped slightly (22.9/54.6), indicating that allowing the grouping mechanism to remain adaptive is important for maintaining accuracy.

Slot Attention, despite being learnable, resulted in a marked drop (19.8/52.2). Investigation revealed that tokens collapsed to the same spatial location across images, even with Sinkhorn normalisation, leading to low-quality groupings. Nonetheless, this setup still outperformed iBOT (18.8/47.2), suggesting that cross-view matching in Phase 3 provides consistent benefits, even when the reference regions are poorly formed.

| Clustering module | ADE | Pascal |
|---|---|---|
| K-means (Frozen) | 23.1 | 54.8 |
| CG-SSL (Frozen clustering Module) | 22.9 | 54.6 |
| Slot-Attention (learnable) | 19.8 | 52.2 |
| **CG-SSL (ours)** | **24.3** | **56.1** |

Table 5: Impact of alternative cluster assignment mechanisms.

## D.3 Number of Concept Tokens $N$

We evaluated the model's robustness to different numbers of concept tokens $N$ as shown in Table 6. Performance remains stable across a range of values, with a peak at $N = 6$. In the main ImageNet experiments, we set $N = 4$ as the dataset is less complex.

| $N$ | ADE | Pascal |
|---|---|---|
| 2 | 23.8 | 55.3 |
| 4 | 24.1 | 55.9 |
| **6** | **24.3** | **56.1** |
| 8 | 24.0 | 55.7 |
| 12 | 23.8 | 55.5 |

Table 6: Sensitivity to the number of concept tokens.

## D.4 Depth $L$ of the Clustering Module

We evaluate the impact of the number of transformer decoder blocks ($L$) in the clustering module. While using $L = 4$ yields the best performance, overall results are relatively stable across different number of blocks.

| $L$ | ADE | Pascal |
|---|---|---|
| 1 | 23.9 | 55.9 |
| 2 | 24.0 | 56.0 |
| **4** | **24.3** | **56.1** |
| 8 | 23.8 | 55.9 |

Table 7: Effect of clustering module depth.

## D.5 Sharing the Projection Head

We investigate whether the projection head should be shared between the [CLS] token and the concept tokens. Empirically, sharing improves performance, suggesting that a unified embedding space may benefit alignment. However, we believe that this design choice requires deeper analysis. For instance, could partial sharing. e.g., sharing early layers while keeping the output branches independent, offer a better trade-off? Should the concept tokens use a lower-dimensional output, given that their role is to capture only concepts, while the [CLS] token must also encode broader context and inter-region relations?

## D.6 Temperature $\kappa$ in Softmax Fusion

We found that a moderate temperature ($\kappa = 0.5$) yields the best balance between sharpness and stability.

| Shared | ADE | Pascal |
|:---:|:---:|:---:|
| ✓ | **24.3** | **56.1** |
| ✗ | 23.4 | 49.8 |

Table 8: Effect of sharing projection head between [CLS] and concept-tokens.

| $\kappa$ | ADE | Pascal |
|:---:|:---:|:---:|
| 0.1 | 23.8 | 55.9 |
| 0.3 | 24.1 | 56.0 |
| **0.5** | **24.3** | **56.1** |
| 0.8 | 23.7 | 55.1 |
| 1.0 | 23.3 | 55.0 |

Table 9: Influence of the temperature parameter in mask fusion.

## D.7 Reconstruction Target

We compare reconstructing raw pixels vs. teacher features. As expected, the latter performs better, which is highlighting the importance of semantically guided reconstruction over low-level signal recovery.

| Target | ADE | Pascal |
|:---|:---:|:---:|
| Pixels (RGB) | 23.8 | 55.6 |
| **Teacher features** | **24.3** | **56.1** |

Table 10: Pixel-space vs. feature-space reconstruction.

## D.8 Retaining the Patch-Level Loss $\mathcal{L}_{\texttt{[patch]}}$

We ablate whether retaining Phase 1's patch-level loss in later stages is beneficial. Removing it results in a modest drop in accuracy, indicating that fine-grained supervision from patch features complements concept-level grouping and alignment.

Separately, we believe that a more systematic investigation into the weighting of individual loss components $\alpha_{1,\ldots,5}$ would be valuable, as it may uncover better trade-offs across the different training objectives.

## D.9 Backbone Scaling

Finally, we verify the robustness of CG-SSL across different backbones, including ViT-S, ViT-B, and ViT-L. Performance scales consistently, with ViT-L achieving the strongest results across both datasets.

| $\mathcal{L}_{\texttt{patch}}$ | ADE | Pascal |
|:---:|:---:|:---:|
| ✓ | **24.3** | **56.1** |
| ✗ | 24.1 | 56.0 |

Table 11: Effect of retaining the Patch-Level Loss $\mathcal{L}_{\texttt{[patch]}}$.

| | ADE | Pascal |
|:---|:---:|:---:|
| ViT-S/16 | 24.3 | 56.1 |
| ViT-B/16 | 26.1 | 58.6 |
| ViT-L/16 | 28.4 | 59.0 |

Table 12: Backbone Scaling.

