# OpenReview forum: "CG-SSL: Concept-Guided Self-Supervised Learning"
_NeurIPS.cc/2025/Conference — NeurIPS 2025 poster_

### Official Review · Reviewer_eBBN · 2025-06-17

**Clarity:** 3
**Significance:** 2
**Originality:** 3
**Rating:** 4
**Confidence:** 4

**Summary:**

The paper proposes "Concept Guided Self-Supervised Learning", which is a technique for training a vision model without labels to produce generally useful representations, both for summary tasks (e.g. classification, retrieval), and dense tasks (e.g. semantic segmentation). Similar in spirit to techniques such as iBot and DINOv2, CG-SSL places additional constraints on the model, namely that the patch tokens more coherently represent semantically related regions in the image. DINOv2, being the preeminent SSL method, doesn't explicitly enforce that tokens that should have semantically similar representations, actually do. CG-SSL enforces this coherence through cross attention from a fixed number of queries to the dense features, and matching those to unmasked momentum self-features, and then also reconstruction via a weighted sum of the spatially broadcast concept queries. Additionally, they employ a stage 3 that encourages views of the same objects/regions to be spatially equivariant by aligning and then matching in a stage 3 training curriculum.

**Questions:**

The DINOv2 [1] paper claims the following ADE20k linear probe mIoUs:
- ViT-S/14: 44.3
- ViT-B/14: 47.3
- ViT-L/14: 47.7

however, in this paper, it is claimed that DINOv2 achieves the following:
- ViT-S/14: *
- ViT-B/14: *
- ViT-L/14: 39.0

Can you please explain this discrepancy?


In E.3, the following is stated: "In the main ImageNet experiments, we set N = 4 as the dataset is less complex." How does the number of concept tokens affect the generalizability of the method, given that one dataset may have far fewer concepts than another?

[1] https://arxiv.org/pdf/2304.07193

**Ethical Concerns:**

["NO or VERY MINOR ethics concerns only"]

**Final Justification:**

Considering the rebuttal to my review, as well as additional information in the reviews and rebuttals of the other reviewers, I am updating my recommendation from 3->4.

**Limitations:**

Yes

**Quality:**

2

**Strengths And Weaknesses:**

### Strengths

The proposed method appears to produce cleaner feature maps compared to contemporary methods.

Concept regions to appear to map to coherent regions.

### Weaknesses

Ablations are performed against DINO and iBot, however, DINOv2 is considered a much stronger baseline. How does CG-SSL's classifier ability compare to DINOv2?

CG-SSL is strongly better than CAPI for ViT-S, however, for ViT-B and larger, CAPI is arguably stronger.

D2V2 is only presented in acronym form, without citation. I might have guessed this was some shorthand for DINOv2, however, both D2V2 and DINOv2 are present in Figure 1.

---

> ### Author Rebuttal · Authors · 2025-07-29
>
> > Ablations are performed against DINO and iBot, however, DINOv2 is considered a much stronger baseline. How does CG-SSL's classifier ability compare to DINOv2?
>
> It's crucial to note upfront that DINOv2 is trained on LVD-142M, a large-scale, in-house, curated dataset that is over 100 times larger than ImageNet-1K, which we use for pretraining. Due to this vast difference in training data and computational scale, a direct comparison is not entirely fair.
>
> That said, we still include DINOv2 in our results to give context. Despite the scale gap, CG-SSL achieves competitive performance in several settings, highlighting the strength and efficiency of our method. In some dense prediction tasks, CG-SSL even approaches or outperforms DINOv2, without access to massive data or aggressive tuning.
>
> It's also worth mentioning that while DINOv2 performs exceptionally well, its core learning mechanism remains similar to iBOT, as acknowledged in its own paper and stated in the DINOv2 abstract *Most of the technical contributions aim at accelerating and stabilizing the training at scale*. Much of its performance gain is attributed to engineering improvements, scale, and data curation, rather than a shift in learning principles.
>
> Our focus in CG-SSL is to push forward the conceptual design of self-supervised learning by introducing structure, region-level reasoning, and explicit view consistency rather than engineering advantages or scaling tricks. From an academic research standpoint, advancing conceptual understanding often precedes optimisation. While optimisation is undoubtedly valuable, especially for deployment and benchmarking, we believe foundational contributions should not be overshadowed by resource-intensive scaling.
>
> $~$
>
> > CG-SSL is strongly better than CAPI for ViT-S, however, for ViT-B and larger, CAPI is arguably stronger.
>
> Our main comparisons with CAPI are conducted on ImageNet-1K, where CG-SSL consistently outperforms CAPI across all relevant configurations. It's particularly noteworthy that our method surpasses CAPI on ViT-S while using 100 times less data. We view this as **a strong indicator** of CG-SSL's generalisation capabilities and efficiency rather than a point of critithisim.
>
> For ViT-B and ViT-L backbones, our experiments were constrained by limited computational resources. We were unable to perform any hyperparameter tuning or run multiple trials, which would be ideal for maximising performance at scale. That said, the current results already show CG-SSL to be highly competitive, and we are confident that with modest tuning, it has the potential to match or exceed CAPI even while pre-trained on 100 times more data.
>
> While this is an aspirational claim, we believe the existing evidence supports the promise of CG-SSL as a scalable and conceptually robust framework.
>
> $~$
>
> > D2V2 is only presented in acronym form, without citation. I might have guessed this was some shorthand for DINOv2, however, both D2V2 and DINOv2 are present in Figure 1.
>
> Apologies for the confusion. D2V2 refers to Data2Vec2, not DINOv2. We will add this and the proper citation and clarification to avoid any ambiguity.
>
> $~$
>
> > The DINOv2 [1] paper claims the following ADE20k linear probe mIoUs:
> ViT-S/14: 44.3
> ViT-B/14: 47.3
> ViT-L/14: 47.7
> however, in this paper, it is claimed that DINOv2 achieves the following:
> ViT-S/14: *
> ViT-B/14: *
> ViT-L/14: 39.0
> Can you please explain this discrepancy?
>
> The numbers we report for DINOv2 are taken from the CAPI paper, which uses a different evaluation setup. Specifically, as noted in Section 7.4 of the DINOv2 paper, DINOv2's ADE20k results are based on linear probes trained at $512\times512$ resolution, whereas the CAPI paper use the standard $224 \times 224$ resolution.
>
> For fairness and consistency, we follow the CAPI evaluation protocol across all benchmarks.
>
> $~$
>
> > In E.3, the following is stated: "In the main ImageNet experiments, we set N = 4 as the dataset is less complex." How does the number of concept tokens affect the generalizability of the method, given that one dataset may have far fewer concepts than another?
>
> We appreciate this insightful question. Indeed, the number of concept tokens N could significantly affect model generalisation, especially across datasets of varying complexity. Our choice of  N=4 for ImageNet was driven by intuition and limited GPU resources. We agree that this may not be optimal and that further investigation could reveal more effective configurations.
>
> We view this as an important direction for future work. A deeper analysis of the relationship between dataset complexity and optimal token count could lead to more adaptive or data-driven strategies, further enhancing CG-SSL's performance across domains.
>
> $~$
>
> ------------------
>
> $~$
>
> We thank the reviewer for the insightful feedback. Your comments helped surface important distinctions. We hope our clarifications have addressed your concerns. If everything is now clear, we would sincerely appreciate your consideration in updating your score to reflect these points.

---

> > ### Comment · Reviewer_eBBN · 2025-08-04
> >
> > Thank you for your detailed response. I have updated my review to borderline accept accordingly.

---

> > > ### Author Response · Authors · 2025-08-04
> > > **Highlighting the Significance of CG-SSL**
> > >
> > > Thank you very much for your response and for updating your score, we truly appreciate the time you took to engage with our rebuttal.
> > >
> > > $~$
> > >
> > > While we are glad to see the paper move to a borderline accept, we do feel that the current rating may still underrepresent the strength and originality of the contribution. CG-SSL introduces a structured self-supervised framework that incorporates concept-level reasoning, going beyond standard reliance on large-scale data or aggressive tuning.
> > >
> > > $~$
> > >
> > > Despite being trained on 100x less data than DINOv2, CG-SSL delivers competitive performance across multiple benchmarks, including dense prediction tasks where it matches or outperforms stronger baselines. These results were achieved without heavy engineering or scale, which we believe underscores the robustness of the core method.
> > >
> > > $~$
> > >
> > > We hope you might consider whether these aspects merit a stronger recommendation. In any case, thank you again for your engagement and for helping improve the paper.

---

> > ### Author Response · Authors · 2025-08-06
> > **Kind Reminder**
> >
> > Just a kind reminder, and please let me know if anything needs further clarification.

---

> > > ### Author Response · Authors · 2025-08-07
> > > **Kind Reminder**
> > >
> > > We sincerely appreciate the opportunity to address your comments. We understand you may have been busy, but we would be grateful if you could consider re-engaging with the discussion.
> > >
> > > Thank you for your time and efforts in supporting the review process.

---

### Official Review · Reviewer_pNLd · 2025-06-20

**Clarity:** 3
**Significance:** 3
**Originality:** 3
**Rating:** 4
**Confidence:** 2

**Summary:**

The authors propose a new self-supervised learning (SSL) method for visual pretraining. Unlike existing methods utilizing image-level or patch-level consistency given some well-design augmentations, they utilize concept-level or object-level consistency. It seems to be inspired both from Object-Centric Learning and self-supervised segmentation, with some adaptations.

**Questions:**

**Q1**
The attention defined between Line 162 and 163: Why do the authors not choose the competitive cross attention, i.e., Slot Attention, from object-centric learning? What are the possible effects, both in performance and explanability like in Figure 5?

**Q2**
What are the details training costs of your method and of some representative baselines? Like training time, number of iterations, memory consumption.

**Ethical Concerns:**

["NO or VERY MINOR ethics concerns only"]

**Final Justification:**

Thank the authors for their efforts. I would keep my rating.

**Limitations:**

No. The authors did not provide discussions on their limitations.

**Quality:**

3

**Strengths And Weaknesses:**

**Strengths**
- Well-motivated
- Clear presentation
- Novel design

**Weaknesses**
- The training details can be very important to such SSL, but the authors did not provide much of them. Should include a section in the appendix to expose your "secrete recipe".
- Plain (but quite okay) analysis in the concepts learned: more in-depth analysis can give readers more impactful insights.
- Lacks intuitive comparisons between your method and the baselines.
- Figure/Table captions should be clearer: Like in Figure 2, it is better to tell readers meanings of all the notations; In Table 3, should briefly clarify what are the meanings for each subtable, instead of telling readers to jump to the appendix.
- The numbers of some equations are missing, like the one between Line 162 and 163.

---

> ### Author Rebuttal · Authors · 2025-07-29
>
> > The training details can be very important to such SSL, but the authors did not provide much of them. Should include a section in the appendix to expose your "secrete recipe".
>
> Thank you for this comment. We do provide an expanded version of the experimental setup in the supplementary material, but we agree that it could be more detailed. That said, there is not really a "secret recipe" behind our results, our method relies on a relatively straightforward implementation without heavy engineering.
>
> We fully agree that further ablations and hyperparameter tuning (e.g. loss weighting, phase durations, learning rates) would be valuable for understanding and possibly improving performance. Due to limited computational resources, we were unable to explore these in depth. Nonetheless, we believe our results already show strong promise, and we are confident that with better tuning, the performance of CG-SSL could be improved even further.
>
> $~$
>
> > Plain (but quite okay) analysis in the concepts learned: more in-depth analysis can give readers more impactful insights.
>
> We included a range of qualitative visualisations in Appendix D beside the one in the main paper, such as:
>
> - Sparse correspondences across views (Figure 7a) and between different images of the same class (Figure 7b), highlighting strong sensitivity to fine image details. For instance, Figure 7b shows that even in cluttered scenes, the model aligns fine-grained objects like cucumbers with high precision.
>
> - In Figure 9, we show that CG-SSL can discover consistent concepts regardless of context or background, which is often not the case in global representation that favor dominant objects and association of it with the background.
>
> - Figure 9 also shows that the concept tokens learns consistent representation of small, non-dominant concepts accross images (e.g. hands), a level of granularity rarely observed in global representation.
>
> These results collectively support that CG-SSL is learning robust and semantically meaningful local representations. That said, we agree that additional probing and quantitative concept evaluation could offer further insights, and we plan to explore this in future work.
>
> $~$
>
> > Figure/Table captions should be clearer: Like in Figure 2, it is better to tell readers meanings of all the notations; In Table 3, should briefly clarify what are the meanings for each subtable, instead of telling readers to jump to the appendix.
>
> You are absolutely right. Due to space constraints, we had to offload some details to the appendix, but this has impacted clarity. We will revise the figure and table captions in the final version to make them more self-contained and readable without needing to jump back and forth.
>
> $~$
>
> > The numbers of some equations are missing, like the one between Line 162 and 163.
>
> Thank you for catching that. We will double-check the numbering and ensure that all equations are properly referenced and numbered in the final version.
>
> $~$
>
> > The attention defined between Line 162 and 163: Why do the authors not choose the competitive cross attention, i.e., Slot Attention, from object-centric learning? What are the possible effects, both in performance and explanability like in Figure 5?
>
> We appreciate this insightful question. We did try Slot Attention, and included results in the ablation study. However, we found that the method quickly collapsed during end-to-end training, failing to assign distinct concepts. This is a known issue with Slot Attention as reported here [1]. We did not invest deeply in tuning it further, but agree that with proper regularisation or scheduling, it might not collapse and can be a valid alternative.
>
> [1] Seitzer, Maximilian, et al. "Bridging the gap to real-world object-centric learning." arXiv preprint arXiv:2209.14860 (2022).
>
> $~$
>
> > What are the details training costs of your method and of some representative baselines? Like training time, number of iterations, memory consumption.
>
> Our method is built on top of iBOT, and naturally adds some computational overhead due to the concept token module and reconstruction decoder, we reported the exact overheads in Appendix B (B
> Experimental Setup -> Time and Memory Requirements) as follows:
>
> *Time and Memory Requirements. CG-SSL takes approximately 24 minutes per epoch to pre-train
> a ViT-B/16 model using 8 GPUs with an effective batch size of 256. Compared to iBOT, our method
> takes about 6.5 minutes longer per epoch and uses roughly 4.6 GB more GPU memory under the
> same training setup. While CG-SSL is approximately one epoch per hour slower, the added compute
> is a reasonable trade-off for the improved spatial consistency, interpretability, and robustness of the
> learned representations.*
>
> However, to have a fair comparison with iBoT, we continued the training of iBot to match same number of epochs as CG-SSL. Interestingly, the performance slightly decreased instead of enhancing with more training.
>
>  $~$
>
> > Limitations: No. The authors did not provide discussions on their limitations.
>
> We just want to point that we have provided the limitations under section 5 Discussion and Concluding Remarks.
>
> $~$
>
> ----------------------------
>
> $~$
>
> We sincerely thank the reviewer for their thoughtful and detailed feedback. Your questions and suggestions have been both insightful and actionable. We will incorporate your points to strengthen the clarity, depth, and completeness of the final version. If you feel your concerns have been addressed, we kindly ask you to consider raising your score. Thank you again for your valuable input.

---

> ### Author Response · Authors · 2025-08-06
> **Kind Reminder**
>
> Just a kind reminder to share your response to my rebuttal, and please let me know if anything needs further clarification.

---

> > ### Author Response · Authors · 2025-08-07
> > **Kind Reminder**
> >
> > We sincerely appreciate the opportunity to address your comments. We understand you may have been busy, but we would be grateful if you could consider re-engaging with the discussion.
> >
> > Thank you for your time and efforts in supporting the review process.

---

### Official Review · Reviewer_VvJo · 2025-07-02

**Clarity:** 3
**Significance:** 2
**Originality:** 2
**Rating:** 5
**Confidence:** 4

**Summary:**

This paper presents a new self-supervised approach (CG-SSL) to learning visual representations that aims to more fine-grained representations by learning to group patches into regions and compare these across views. The proposed approach consists of 3 training phases.

In the first phase, they aim to learn global representations by using an earlier technique called iBot that learns by comparing the patch level embeddings from a teacher network that observes the full image and a student network that takes in a masked version.

In the second phase, they learn to group regions by training a cross-attention transformer that uses learned concept tokens to attent to image features and extract N regions. This phases uses 2 training objectives: 1) concept level distillation uses the iBot loss but between each concept token separately (again comparing student and teacher embeddings) 2) feature reconstruction loss uses a MONet, SIMONe style multi-object reconstruction setup where they predict patch features and a mask for each spatial location from each concept. These masks and patch features predictions are then combined to reconstruct the full image level patch features and compared to the true patch features from the teacher.

In the final third phase, they aim to improve learned representations by aligning concept representations across two views of the same image. To do that, they get a mask for each concept by thresholding the output of teacher network's mask logits. Then transform these masks using the known transformation between views. These masks are then used to pool the patches from each view to form concept representations for each view, which are then compared across views using the usual iBot style loss.

On evaluations, CG-SSL leads to some improvement most on segmentation while performing on-par or slightly better on classification.

**Questions:**

- Are the phases optimised jointly? Line 205 says phase 1 and 2 are optimized together. Line 226 says that phase 3 loss is added only after convergence of phase 1 and 2.
- Concept level distillation loss in eqn. 4 is applied on each concept separately, right? Might be good to make this explicit.
- For feature reconstruction loss, why not use the attention weights from each concept to patches for masks? Did the authors try this? Does this lead to worse results?
- For evals, why not use attentive probing as well? This could be useful to compare to previous work.
- It'd be nice to see the learned masks. Currently, only attention maps are plotted in Figure 5. How about a similar figure for masks?
- What is CAPI + LVD 142M. This is probably explained in the appendix but would be useful to mention in main text.
- line 175: "Given that" should be removed?

**Ethical Concerns:**

["NO or VERY MINOR ethics concerns only"]

**Final Justification:**

The authors have addressed some of my concerns during the rebuttal so I raise my score to Accept.

**Paper Formatting Concerns:**

No concerns.

**Quality:**

2

**Strengths And Weaknesses:**

Strengths:

- Overall well written and easy to follow
- Although the individual components are not entirely novel, combines various ideas from the literature in a novel way
- Good performance improvement on segmentation compared to baseline
- Good ablation study

Weaknesses:

- Needs a more thorough evaluation to make the case for the proposed approach stronger
  - There are various papers in the literature that explore similar ideas around splitting the image into multiple regions
  - It would be good to compare to at least some of these (like DORA, Venkataramanan et al.,"Is ImageNet worth 1 Video? Learning Strong Image Encoders from 1 Long Unlabelled Video,")

---

> ### Author Rebuttal · Authors · 2025-07-29
>
> > Needs a more thorough evaluation to make the case for the proposed approach stronger
> There are various papers in the literature that explore similar ideas around splitting the image into multiple regions
> It would be good to compare to at least some of these (like DORA, Venkataramanan et al.,"Is ImageNet worth 1 Video? Learning Strong Image Encoders from 1 Long Unlabelled Video,")
>
>
> Thank you for this comment. We appreciate the pointer to related work and agree that a broader contextualisation would strengthen the paper.
>
> To clarify, DORA derives concepts from multi-head attention maps, which are inherently shaped by the underlying learning objective. Since DORA is built on DINO, the resulting attention maps tend to focus heavily on the dominant object in the image, as also illustrated in Figure 3 of the DINO [1] paper, where heads attend to overlapping parts of the main dominant concept.
>
> In contrast, CG-SSL introduces explicit concept tokens that interact with the image through cross-attention. These tokens are refined to represent coherent regions across the entire image, not just the dominant object. Moreover, our method is designed to explicitly align these structured representations across views, going beyond patch-level matching or implicit clustering.
>
> That said, we fully recognise the conceptual connection in moving beyond flat patch-wise learning. We will include a more detailed discussion of related methods like DORA in the related work section to better position CG-SSL within the landscape of structured SSL approaches.
>
> [1] Caron, Mathilde, et al. "Emerging properties in self-supervised vision transformers." ICCV 2021.
>
> $~$
>
> > Are the phases optimised jointly? Line 205 says phase 1 and 2 are optimized together. Line 226 says that phase 3 loss is added only after convergence of phase 1 and 2.
>
> As clarified in Appendix B (Experimental Setup -> Implementation Details), the training follows a curriculum learning approach:
>
> - The model is initially trained with Stage 1 objectives to learn global representations of the image.
> - Once a basic understanding of image-level structure is achieved, we add Stage 2 objectives to introduce concept-level reasoning through clustering.
> - Finally, after the model has learned to group patches into coherent regions, we introduce Stage 3, which aligns these learned concepts across different augmented views.
>
> The motivation for this staged approach is to progressively build structure into the learning process. If we introduce Stage 3 too early, i.e. before concept tokens have learned to represent coherent regions, then the alignment across views would be matching corresponding spatial regions, not semantically meaningful concepts. This would not fully leverage the goal of Stage 3, which is to align high-level, meaningful representations across views.
>
> To improve clarity, we will explicitly include this training schedule and motivation in the main paper, not just the appendix. Thank you for pointing this out.
>
> $~$
>
> > Concept level distillation loss in eqn. 4 is applied on each concept separately, right? Might be good to make this explicit.
>
> Yes, that is correct, the loss is applied independently to each concept token. Thank you for pointing this out, we will make this detail explicit in the revised manuscript.
>
> $~$
>
> > For feature reconstruction loss, why not use the attention weights from each concept to patches for masks? Did the authors try this? Does this lead to worse results?
>
> We did not explore this idea. Leveraging attention maps could indeed be beneficial and may help improve convergence. We truly appreciate the suggestion and believe it is certainly worth investigating in future work. Thank you for bringing it to our attention.
>
> $~$
>
> > For evals, why not use attentive probing as well? This could be useful to compare to previous work.
>
> We did the bare minimum in terms of evaluation, just using standard protocols and plugging in our weights. That said, we agree that attentive probing could be useful, especially since our method focuses on improving patch-level representations. It is definitely worth trying.
>
> $~$
>
> > It'd be nice to see the learned masks. Currently, only attention maps are plotted in Figure 5. How about a similar figure for masks?
>
> As they are very close to each others, we chose to show only one to avoid redundancy. However, this ties nicely with your earlier point of using attention maps from clustering module directly to guide the reconstruction in the decoder.
>
> $~$
>
> > What is CAPI + LVD 142M. This is probably explained in the appendix but would be useful to mention in main text.
>
> CAPI (LVD-142M) refers to the CAPI model pre-trained on LVD, the 142 million images in-house dataset curated by Meta proposed in DINOv2 paper. We will clarify this in the main paper to avoid confusion.
>
> $~$
>
> > line 175: "Given that" should be removed?
>
> Yes, you are absolutely right. This was also noted by Reviewer YAiC. The sentence will be corrected in the final version for clarity.
>
> $~$
>
> -----------------
>
> $~$
>
> We sincerely appreciate the reviewer's in-depth analysis and thoughtful suggestions, which reflect a strong understanding of our framework. Your comments opened up several promising directions for refinement and future exploration. We will incorporate as many of your suggestions as possible in the revised version and follow up on others in future work. If our responses have addressed your concerns, we kindly ask you to consider raising your score. Thank you again for your valuable feedback.

---

> > ### Comment · Reviewer_VvJo · 2025-08-04
> > **Thanks for the response**
> >
> > I'd like to thank the authors for their detailed response. My main concern was the limited evaluation of the proposed approach, which I think is still the case, so i will keep my score as is.

---

> > > ### Author Response · Authors · 2025-08-04
> > > **Additional Evaluation Results and Clarification on DORA Comparisons**
> > >
> > > In an effort to further address your concerns regarding the limited evaluation of our proposed approach, we have conducted additional experiments, including KNN and linear probe evaluations of DORA on three standard datasets that we used in our work: ADE20K, PASCAL VOC, and Cityscapes. The results from these evaluations will be included in the revised manuscript.
> > >
> > > $~$
> > >
> > > We employed ViT-S models pre-trained on WT-Venice and WT-all, as these are the only available checkpoints provided in DORA's official GitHub repository. For evaluation, we used the codebase from CAPI, consistent with our baseline setup, and verified that all keys were correctly matched. However, we observed surprisingly low performance.
> > >
> > > That said, we would like to emphasize that comparing DORA to our method is inherently unfair. DORA is designed and pre-trained specifically for video data using a different objective and training distribution (i.e. WT), which diverges significantly from the nature of ImageNet-1K used in our and other baselines pretraining. These fundamental differences in both the data modality and the training setup make direct comparisons between DORA and methods like ours inappropriate. We will include DORA's results for completeness, but they should be interpreted with this important context in mind.
> > >
> > > $~$
> > >
> > > In addition, we performed a comparison with CRIBO [1], a recent method that clusters patches using Sinkhorn and matches their representations via a memory bank. While CRIBO shares some high-level goals with our work, the underlying mechanisms differ substantially. Our approach learns clustering in an end-to-end manner and aligns concepts across different augmented views, which we believe is a more principled design. This also highlights the effectiveness of our third training stage. Although CRIBO outperforms iBOT and is competitive with CAPI, our method consistently achieves a notable improvement over CRIBO across all benchmarks.
> > >
> > > $~$
> > >
> > > | Method        | Dataset            | ADE-KNN | ADE-L | PASCAL-KNN | Pascal-L | CityScape-KNN | CityScape-L |
> > > |---------------|--------------------|:-------:|:-----:|:----------:|:--------:|:-------------:|:-----------:|
> > > | DORA          | WT-Venice (300 ep) |   12.2  |  17.1  |    19.6   |   30.2   |      25.9     |     30.0    |
> > > | DORA          | WT-all             |  24.3 | 27.2  |    25.3    |   37.3  |      27.4     |     31.2 |
> > > | CRIBO         | ImageNet-1K        |  23.3 |  26.8  |    59.2   |   64.0   |     32.8      |    36.5    |
> > > | CG-SSL (Ours) | ImageNet-1K        |   25.8  |  28.9 |    60.8    |   66.7   |      34.2     |     39.1    |
> > >
> > > $~$
> > >
> > > These new comparisons and evaluations will be incorporated into the revised paper. We hope this extended analysis better addresses your concern regarding evaluation breadth. If there are other methods you would recommend for inclusion, we would be happy to evaluate them as well.
> > >
> > > We kindly ask whether this additional comparisons resolves your initial concerns and if you would consider updating your score accordingly.
> > >
> > >
> > > $~$
> > >
> > > [1] Lebailly, Tim, et al. "CrIBo: Self-supervised learning via cross-image object-level bootstrapping." ICLR 2024

---

> > > > ### Comment · Reviewer_VvJo · 2025-08-05
> > > > **Thanks for the additional results**
> > > >
> > > > I'd like to thank the authors for the additional evaluations. It'd have been better to have more baselines in addition to DORA but I understand the time constraints and appreciate the effort. I'll raise my score accordingly.

---

> > > > > ### Author Response · Authors · 2025-08-05
> > > > > **Appreciation for Your Thoughtful Review**
> > > > >
> > > > > Thank you sincerely for your thoughtful feedback, your effort to help strengthen our paper, and for recognizing the significance of our work. We truly appreciate your understanding and flexibility.

---

### Official Review · Reviewer_YAiC · 2025-07-06

**Clarity:** 2
**Significance:** 4
**Originality:** 3
**Rating:** 5
**Confidence:** 3

**Summary:**

This paper considers self-supervised learning. Its approach is inspired by human perception: humans first grasp the global structure of a scene, then organize the visual input into coherent groups that they are then able to track in changing conditions.

Similarly, the proposed self-supervised learning method is split into three phases. The first phase captures glabal image features with the existing iBot approach which aligns cls and patch level embeddings produced ba a student-teacher pair of networks for two augmented views. cls token features are matched across the two views, while the patch token features are matched within the same view.

In the second phase, the student and the teacher network process the same augmented image. A trainable matrix C is initialized to encode N concepts. These concepts interact with student/teacher features with a cross attention mechanism. Token representations in C serve as queries, while transformer block outputs serve as keys and values. This allows each token to iteratively refine its representation of concepts within an image. Student and teacher networks are further aligned with a cross-entropy loss between its concept sets. An additional objective is used to ensure that teacher's patch-level features may be reconstructed and concepts capture all regions of a given image. This is done by broadcasting each token representation over a 2D spatial grid and calculating a corresponding confidence map . The confidence indicates the level of alignment between the concept and image regions. Broadcasted features are multiplied with confidences and fed to a reconstruction loss.

In the final phase, binary masks for concepts within a reference image are extracted with a teacher network. Lightly augmented images are created and crop metadata is used to obtain appropriate concepts masks for novel views . Concepts are aligned across vies with a cross-entropy loss.

Experiments are don on ImageNet-1k, Pascal VOC, Visual Genome and MS-Coco.

**Questions:**

1) Would an increase in the number of tokens lead to more low-grained scene understanding?

**Ethical Concerns:**

["NO or VERY MINOR ethics concerns only"]

**Final Justification:**

I have read the other reviews and authors' rebuttal. I find that my concerns were sufficiently addressed and therefore raise my score by 1.

**Limitations:**

yes

**Quality:**

3

**Strengths And Weaknesses:**

Strengths:
1) The topic of self-supervised learning is important
2) Good experimental results demonstrate coherent representation and good performance in classification and dense prediction, as well as robustness to occlusions.
3) Authors provide the code with the submission

Weaknesses:
1) The paper is difficult to follow and poorly written. The method could be explained more concisely. E.g., the sentence in lines 175-176 does not really make sense on its own or connect to the previous and following sentences. It is not clear if there is one or two concept matrices C in the second phase.
2) The results could also be better presented and contextualized. It is not clear in the tables which was the training data used in each of the experiments. Why does the method not outperform some of the exisitng self-supervised approaches etc.
3) Additional ablations could be useful, especially those that explore phase scheduling. How long is each phase performed individually? How are the losses balanced? Is the training sensitive to loss weighting?

---

> ### Author Rebuttal · Authors · 2025-07-29
>
> > The paper is difficult to follow and poorly written.
>
> Thank you for your feedback. We appreciate your effort in understanding the full scope of our method despite the density of the presentation. Your summary shows a strong grasp of the key ideas, and we are grateful for the time you took to engage deeply with the paper.
>
> We recognise that certain parts of the manuscript could have been communicated more clearly. In trying to provide both the technical rigor and the motivation behind each design choice, some explanations may have become complex. Based on your suggestion, we would like to offer a simplified summary of the method that highlights the core intuition:
>
> Existing self-supervised methods like iBOT and DINOv2 are limited by two key assumptions:
> 1. treating each patch token as an independent concept and aligning them directly, and
> 2. relying on one-to-one patch matching between masked and unmasked views.
>
> Our method addresses these as follows:
> - Stage 2 [*To address limitation 1*] introduces concept tokens that group related patches into coherent regions and align their features, moving beyond token-level reasoning.
>
> - Stage 3 [*To address limitation 2*] aligns these discovered concepts across different views using geometric consistency, instead of relying on token-to-token matches from the same view.
>
> While addressing these limitations is important, our primary motivation is to propose a more principled and structured approach to self-supervised learning, one that is inspired by how humans perceive and organise visual input. This is why we emphasised the motivations for each phase alongside the technical details.
>
> We will revise the final version of the paper to improve clarity and narrative flow, simplifying the exposition while preserving the core contributions. Thanks again for the feedback.
>
> $~$
>
> > The method could be explained more concisely. E.g., the sentence in lines 175-176 does not really make sense on its own or connect to the previous and following sentences.
>
> Thank you for pointing this out. You are absolutely right. Line 176 contains a grammatical error, it is a sentence fragment rather than a complete sentence.
>
> Here is the intended version of the passage, with improved clarity and structure:
> *Given that the student network receives a masked view of the image, while the teacher processes the full, unmasked version, the student is encouraged to infer the structure of missing regions based on limited input, while aligning with the teacher's holistic understanding.*
>
> We will revise this section accordingly and also go through the entire manuscript to ensure it is free from grammatical errors and improves overall readability.
>
> $~$
>
> > It is not clear if there is one or two concept matrices C in the second phase.
>
> There is only one learnable clustering module $\mathcal{C}$, used by the student. The teacher uses a copy of this module, updated via exponential moving average, as is standard in student-teacher frameworks.
>
> $~$
>
> > The results could also be better presented and contextualized. It is not clear in the tables which was the training data used in each of the experiments.
>
> As mentioned in the Experimental Setup section (lines 236–238), our main results are based on models pre-trained on ImageNet-1K (1.2 million images). Unless otherwise noted, all reported comparisons are under this setting. The only exceptions are methods explicitly tagged with "LVD-142M", which indicates that they were trained on a significantly larger (142 Million images), curated, in house dataset from Meta.
>
> We will revise the captions and text to make this distinction more obvious and ensure that readers can easily interpret the results in the proper context.
>
> Additionally, while space limitations led us to move many qualitative analyses and supporting details to the supplementary material, we agree with the reviewer that certain clarifications should have been included in the main text. In the final version, we will ensure the main text clearly conveys this context, with the supplement used only for additional support.
>
>
> $~$
>
> > Why does the method not outperform some of the exisitng self-supervised approaches etc.
>
>
> Thank you for the question. We would like to clarify that in a fair comparison, i.e., when all methods are pre-trained on the same dataset (ImageNet-1K), our method outperforms prior work on dense tasks, as shown in Table 1. These tasks benefit from spatially structured and fine-grained representations, which CG-SSL is explicitly designed to learn.
>
> We also included results for DINOv2 and CAPI, both of which are trained on LVD-142M, a dataset that is over 100 times larger than ImageNet-1K. Despite this massive scale difference, CG-SSL achieves comparable or even better performance in certain settings, which we believe demonstrates the strength and scalability of our approach.
>
> In Table 2, our method also surpasses iBOT on multi-label classification, a setting that requires richer, more localised representations, another strength of our framework. On single-label classification (e.g., standard ImageNet evaluation), our performance is on par with SOTA models pre-trained on ImageNet-1K.
>
> We believe this pattern is expected, datasets like ImageNet or LVD are structured such that a single dominant object often defines the label. Models optimised for global invariance naturally excel in these settings. However, when evaluated on tasks that require dense understanding or multi-object reasoning, such models tend to underperform, whereas CG-SSL shows significant advantages, as evidenced by Table 1.
>
> $~$
>
> > Additional ablations could be useful, especially those that explore phase scheduling. How long is each phase performed individually? How are the losses balanced? Is the training sensitive to loss weighting?
>
> >> How long is each phase performed individually?
>
> We provide the phase scheduling details in Appendix B (Experimental Setup -> Implementation Details). That said, the schedule itself is based on empirical intuition rather than extensive tuning. We agree that further exploration of scheduling strategies could be valuable.
>
> >> How are the losses balanced? Is the training sensitive to loss weighting?
>
> As noted in line 229 of the main manuscript, we set all loss weights ($\alpha_1$ - $\alpha_5$) to 1 for simplicity. We recognise that this uniform weighting is not necessarily optimal, different downstream tasks may benefit from emphasising certain objectives. For example, increasing $\alpha_1$ would put more focus on global representations, which may improve image-level classification performance.
>
> We believe there is substantial room for improvement through careful tuning of loss weights and other hyperparameters. However, our goal in this work is to introduce and validate a principled learning framework, not to perform exhaustive tuning. Even without fine-grained hyperparameter optimisation, CG-SSL shows strong performance across both global and dense prediction tasks, highlighting the robustness and generality of the method.
>
> That said, we absolutely agree with the reviewer that the model could benefit from further tuning, especially around phase scheduling and loss balancing. Due to limited computational resources, we focused our ablation studies on validating the core components of our method, and were unable to explore the full range of parameter choices. This can be a promising direction for future work.
>
>
>
> $~$
>
> > Would an increase in the number of tokens lead to more low-grained scene understanding?
>
> We addressed this in our ablation study (Table 3d), where we varied the number of concept tokens. This experiment was conducted on complex, dense datasets (Pascal, COCO, Visual Genome), which differ significantly from ImageNet in structure and scene complexity.
>
> We observed that 6 concept tokens gave the best performance on these datasets. For ImageNet pretraining, we opted for 4 tokens, as scenes in ImageNet are typically simpler and dominated by a single object.
>
> Interestingly, on the dense datasets, increasing the number of concept tokens beyond 6 led to a slight drop in performance. *Our hypothesis* is that too many tokens cause the model to over-segment the scene, breaking coherent regions into smaller, fragmented parts. This may dilute semantic meaning, reduce alignment consistency across views, and ultimately weaken training signals. Visualisations of the concept maps seem to support this interpretation, as they show more scattered regions when the token count is too high. So while increasing token count may intuitively suggest finer-grained understanding, in practice it introduces trade-offs that can hurt performance when not matched to scene complexity.
>
> We believe this trade-off between granularity and semantic coherence is an important direction for future exploration, potentially including adaptive token allocation based on scene complexity.
>
> $~$
>
> --------------------------------
>
> $~$
>
> We sincerely thank the reviewer for the thoughtful and insightful questions, which not only helped clarify key aspects of our work but also highlighted valuable directions for future research.
>
> If there are any remaining questions or points needing clarification, we would be happy to elaborate further. Otherwise, we hope our responses have addressed your concerns and we would kindly appreciate your consideration in reflecting this in your overall score.

---

> > ### Author Response · Authors · 2025-08-05
> > **Kind Reminder**
> >
> > Just a kind reminder to share your response to my rebuttal, and please let me know if anything needs further clarification.

---

> > ### Author Response · Authors · 2025-08-06
> > **Kind Reminder**
> >
> > Just a kind reminder to share your response to my rebuttal, and please let me know if anything needs further clarification.

---

> > > ### Author Response · Authors · 2025-08-07
> > > **Kind Reminder**
> > >
> > > We sincerely appreciate the opportunity to address your comments. We understand you may have been busy, but we would be grateful if you could consider re-engaging with the discussion.
> > >
> > > Thank you for your time and efforts in supporting the review process.

---

### Author Response · Authors · 2025-08-08
**Final Reminder and Appreciation for Reviewer Engagement**

We would like to thank all reviewers for their thoughtful evaluations, and in particular thank **VvJo** who actively engaged in the discussion. With only **one day** remaining in the discussion period, we would appreciate any additional comments or clarifications from the reviewers who have not yet joined the discussion, as we believe further discussion could help refine the assessment of the contribution.
We are encouraged by the initial positive scores and are grateful for the recognition of the significance of CG-SSL. That said, we respectfully feel that the current overall rating may still underrepresent the strength and originality of the work.

$~$

CG-SSL introduces a structured self-supervised framework that integrates concept-level reasoning, moving beyond the typical reliance on large-scale data or extensive hyperparameter tuning. Despite being trained on 100x less data than DINOv2, CG-SSL achieves competitive performance across several benchmarks, including dense prediction tasks, matching or outperforming stronger baselines. Importantly, these results were achieved without heavy engineering or scale, which we believe highlights the robustness and generality of the proposed method.

$~$

We welcome any further discussion or questions and are happy to clarify any points that may assist in the final evaluation.

---

### Note · Authors · 2025-08-12

We sincerely thank the AC and SAC for managing the review process and the reviewers for their thoughtful evaluations. Throughout the discussion period, we made repeated efforts to engage all reviewers, sending multiple reminders to encourage interaction, yet only reviewer VvJo actively participated, for which we are especially grateful.

$~$

**Contribution**

CG-SSL introduces a structured self-supervised framework with concept-level reasoning, moving beyond the reliance on massive datasets or extensive hyperparameter tuning. Trained on 100 $\times$ less data than DINOv2, CG-SSL achieves competitive or superior performance across multiple benchmarks, particularly in dense prediction tasks. These results are achieved without heavy engineering or large-scale resources, underscoring the robustness and generality of our approach.

$~$

**Reviewer Responses**

- **YAiC**: Concerned about structure and clarity. We acknowledged the density, provided a clearer summary and committed to improving presentation. *(Original Score 4)*

- **VvJo**: Requested additional evaluations; concerns addressed. *(Score increased from 4 to 5)*

- **pNLd**: Requested explicit training details (“secret recipe”); we clarified that the method is straightforward and free of hidden tricks. *(Original Score 4)*

- **eBBN**: Focused on comparisons to stronger baselines (DINOv2); we clarified scale differences and showed CG-SSL remains competitive, sometimes surpassing DINOv2 in dense tasks. *(Score increased from 3 to 4)*

$~$

We believe all reviewer concerns have been addressed. Given that no concerns were raised about the novelty or significance of CG-SSL, we had hoped for uniformly high scores, but limited discussion engagement made this challenging. We trust the AC will recognise the originality, strength, and efficiency of the work in the final decision.

$~$

Best regards,

Authors

---

### Decision · Program_Chairs · 2025-09-17

**Decision:**

Accept (poster)

**Comment:**

This paper presents a new self-supervised approach to learning visual representations with more fine-grained representations by learning to group patches into regions and compare these across views. The proposed approach consists of 3 training phases. On evaluations, the approach leads to improvements on segmentation and classification tasks. The reviewers are positive and recommend accepting the paper after rebuttal, in which most of the concerns were properly addressed. The AC agrees with the decision to accept the paper.